# Origin of band gaps in 3$d$ perovskite oxides

Julien Varignon[1], Manuel Bibes [1] & Alex Zunger[2]

With their broad range of properties, $ABO_3$ transition metal perovskite oxides have long served as a platform for device applications and as a testing bed for different condensed matter theories. Their insulating character and structural distortions are often ascribed to dynamical electronic correlations within a universal, symmetry-conserving paradigm. This view restricts predictive theory to complex computational schemes, going beyond density functional theory (DFT). Here, we show that, if one allows symmetry-breaking energy-lowering crystal symmetry reductions and electronic instabilities within DFT, one successfully and systematically recovers the trends in the observed band gaps, magnetic moments, type of magnetic and crystallographic ground state, bond disproportionation and ligand hole effects, Mott vs. charge transfer insulator behaviors, and the amplitude of structural deformation modes including Jahn-Teller in low temperature spin-ordered and high temperature disordered paramagnetic phases. We then provide a classification of the four mechanisms of gap formation and establish DFT as a reliable base platform to study the ground state properties in complex oxides.

[1] Unité Mixte de Physique, CNRS/Thales, Université Paris Sud, Université Paris-Saclay, 91767 Palaiseau, France. [2] University of Colorado Boulder Colorado, Boulder, CO 80309, USA. Correspondence and requests for materials should be addressed to A.Z. (email: alex.zunger@colorado.edu)

The striking range of experimentally observed low-temperature (LT) and high-temperature (HT) magnetic, electronic, and structural properties across the 3d $ABO_3$ perovskites family have held the condensed matter physics community in constant fascination for many years[1–4]. Regarding magnetism, the LT phases are generally spin-ordered, either ferromagnetic (FM) (e.g., $YTiO_3$[5]) or antiferromagnetic (AFM) (e.g., $LaVO_3$[6], $CaMnO_3$[7], $LaMnO_3$[8], $CaFeO_3$[9], $LaFeO_3$[8], or $YNiO_3$[10]), whereas the HT phases exhibit spin-disordered para-magnetism (PM). Regarding the electronic metal vs. insulator bandgap characteristics, these compounds show three modalities: (i) most LT and HT phases are insulating, except (ii) $CaVO_3$ and $SrVO_3$ that are PM metals at all temperatures[11], whereas (iii) $CaFeO_3$[9] or $YNiO_3$[10,12] display both PM metal and PM insulator HT phases. Regarding structural aspects, these per-ovskites show a range of crystal structure types (cubic, mono-clinic, and orthorhombic) as well as structural distortions within such space groups (including octahedral rotations, anti-polar displacements), as well as Jahn–Teller ($LaVO_3$, $LaMnO_3$) or breathing distortion ($CaFeO_3$ and $YNiO_3$) types (see Supplementary Fig. 1). The HT PM phases usually inherit the LT structure, although they can also transform to their own, distinct structure type (as in $LaVO_3$, $CaFeO_3$, and $YNiO_3$).

The enormous number of publications in this field typically focus on one or just few selected $ABO_3$ compound(s) and one or two of the effects noted above. This makes it difficult to assess the key question: what is the minimal physical description needed to capture the basic magnetic, transport, and structural ground-state properties, and can one define a single, overarching theoretical framework that works essentially across the board? This question is highly consequential because the $ABO_3$ system of compounds is in constant demand for theoretical support in the areas of applications that depend on gapping, such as cata-lysis, water splitting, transparent conductors, thermoelectricity, piezoelectricity, ferroelectricity, and heterostructures showing two-dimensional electron gases (2DEG)[13].

The question of the origin of Mott bandgaps is central to this field: Perhaps the most celebrated issue in this regard, raised by Peierls and Mott[3,4], concerns the way in which a bandgap can form in the presence of an odd number of electrons, when the Fermi energy $E_F$ would intersect what in basic band theory would appear to be a partially occupied, gapless band[3]. The classic Mott–Hubbard "strongly correlated" symmetry-conserving view has formed a central paradigm to resolve this apparent incon-sistency and for teaching and explaining the known phenomen-ology[14–16]. This viewpoint was motivated by the fact that in the extreme ionic limit (where the 3d ion carries all of the active electrons, while $O^{2-}$ is a rigid, closed shell), the energy levels near $E_F$ are degenerate with partial occupancy (such as the triply degenerate $t_{2g}$ occupied by just two electrons, or the doubly degenerate $e_g$ occupied by just a single electron). The question that arose was whether lifting such degeneracies, as needed for gapping, would require specialized correlated methodologies that go beyond what band theory would accommodate. The Mott–Hubbard mechanism for gap formation and magnetism in such d-electron oxides (Fig. 1a) envisions electrons moving across the lattice, forming atomic-like states on certain 3d atomic sites with doubly occupied d orbitals ("upper Hubbard band") and empty d orbitals on other sites ("lower Hubbard band"). In this picture, the bandgap of the AFM and PM phases of these oxides therefore emerges due to this correlation-induced electron–electron repulsion (Fig. 1b top panel), without spatial symmetry breaking (such as structural distortions or magnetic order that would lower the symmetry of the degenerate states). In this view, symmetry can break afterward, but is nonessential for gapping. From this point of view, the minimal theoretical

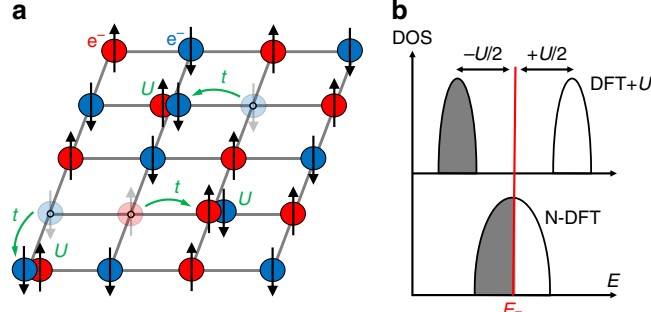

**Fig. 1** Role of the interelectronic $U$ parameter. **a** Mott–Hubbard mechanism with one d electron per site having either a spin up or a spin down. Assuming no symmetry breaking, electrons are hopping on the lattice with a probability $t$ and produce doubly occupied (upper Hubbard band) and empty (lower Hubbard band) orbitals on the sites. Double occupancy causes energy penalty $U$. **b** The naive density functional theory (N-DFT, lower panel) also assuming no symmetry breaking would have instead an ungapped, metallic band

description needed would entail multi-determinant dynamically correlated methodologies capable of treating open-shell degen-erate configurations. Band theory (depicted schematically in the lower part of Fig. 1b) was imagined to fail in producing the observed gaps and structures. Note that in most dynamically correlated calculations, the crystal structure and its subtle dis-tortions were not predicted from the underlying electronic structure (with the exception of the works of Marianetti et al.[17] and Leonov et al.[18] for instance) but rather copied from experi-mentally observed structures. Thus, these simulations are not a proof of the dominant role of dynamical electron correlations over structural symmetry breaking modes for the gap opening. This question is open.

However, the failure of naive DFT models does not disqualify DFT. The question is: what type of band theory fails so drama-tically to describe gapping in 3d compounds (lower part of Fig. 1b). A natural adoption of this symmetry-conserving prin-ciple within band theory for PM phases is to assume a structure where all 3d ions are symmetry-equivalent, having identical local environments (a monomorphous representation). However, since the total magnetic moment in the PM phase must be zero, such a nonmagnetic description forces zero moment on an atom-by-atom basis. This is bound to produce zero gap in a band-theoretic description for systems with an odd number of electrons, in stark contrast with experiments for most AFM and PM phases (Fig. 2).

A common-sense examination of the reasonableness of this approach, apparently missed earlier, is to compare the total energy of this nonmagnetic (NM) approximation with the total energy of the same DFT, but using a proper spin-polarized ground state. We find that the naive DFT has a higher total energy of 452, 1167, 935, 4313, and 1925 meV/f.u for $YTiO_3$, $LaVO_3$, $CaMnO_3$, $LaMnO_3$, and $CaFeO_3$, respectively (see Fig. 2 and Supplementary Note 2 for further details). In other words, the ansatz of a NM state is a highly unlikely scenario.

Even though naive DFT approximations correspond to extre-mely high-energy solutions, they were often used in the literature to suggest that DFT band theory fails to explain gapping of Mott insulators, the latter being argued to require instead an explicitly correlated approach[18–22]. In the specific case of rare-earth nick-elates, it was stated that "Standard DFT and DFT + U methods fail to describe the phase diagram, with DFT predicting that all compounds remain metallic and undisproportionated…. These results establish that strong electronic correlations are crucial to structural phase stability and methods beyond DFT and DFT + U

| | | $CaMnO_3$ | $LaFeO_3$ | $YTiO_3$ | $CaVO_3$ | $SrVO_3$ | $LaMnO_3$ | $LaVO_3$ | $CaFeO_3$ | $YNiO_3$ |
|---|---|---|---|---|---|---|---|---|---|---|
| | | $t_{2g}^3 e_g^0$ | $t_{2g}^3 e_g^2$ | $t_{2g}^1 e_g^0$ | $t_{2g}^1 e_g^0$ | $t_{2g}^1 e_g^0$ | $t_{2g}^3 e_g^1$ | $t_{2g}^2 e_g^0$ | $t_{2g}^3 e_g^1$ | $t_{2g}^6 e_g^1$ |
| | $t$ factor | 0.98 | 0.94 | 0.87 | 0.96 | 1.02 | 0.94 | 0.89 | 0.89 | 0.92 |
| | Gapping mechanism | Octahedral crystal field splitting | | Symmetry lowering | | | | Symmetry lowering + Jahn-Teller effect | Disproportionation effects | |
| **LT phase** | Structure | Pbnm ✓ | Pbnm ✓ | Pbnm ✓ | Pbnm ✓ | Pm-3m ✓ | Pbnm ✓ | $P2_1/b$ ✓ | $P2_1/n$ ✓ | $P2_1/n$ ✓ |
| | Magnetism | AFM-G ✓ | AFM-G ✓ | FM ✓ | PM ✓ | PM ✓ | AFM-A ✓ | AFM-C ✓ | Spiral ✗ (FM) | AFM-S ✓ |
| | Met-Ins | Ins. ✓ | Ins. ✓ | Ins. ✓ | Metal ✓ | Metal ✓ | Ins. ✓ | Ins. ✓ | Ins. ✓ | Ins. ✓ |
| | $E_g$ (eV)/m ($\mu_B$) | 0.69 / 2.49 | 2.31 / 4.12 | 1.11 / 0.94 | 0 / 0.79 | 0 / 0.78 | 1.86 / 3.81 | 1.73 / 1.91 | 0.11 / 3.85-3.42 | 0.50 / 1.19-0.00 |
| | $\Delta E_{LT-HT}$ (meV/f.u) | -23 | -118 | -10 | 0 | 0 | -4 | -10 | -52 | -4 |
| **HT PM phase** | Structure | Pbnm ✓ (NM: ✓) | Pbnm ✓ (NM: ✓) | Pbnm ✓ (NM: ✓) | Pbnm ✓ (NM: ✓) | Pm-3m ✓ (NM: ✓) | Pbnm ✓ (NM: ✓) | Pbnm ✓ (NM: ✗) | 1: $P2_1/n$ ✓ / 2: Pbnm ✓ (NM 1: ✗) | 1: $P2_1/n$ ✓ / 2: Pbnm ✓ (NM 1: ✗) |
| | Met-Ins | Ins. ✓ (NM: ✗) | Ins. ✓ (NM: ✗) | Ins. ✓ (NM: ✗) | Metal ✓ (NM: ✓) | Metal ✓ (NM: ✓) | Ins. ✓ (NM: ✗) | Ins. ✓ (NM: ✓) | 1: Ins. ✓ / 2: Metal ✓ (NM 1: ✗) | 1: Ins. ✓ / 2: Metal ✓ (NM 1: ✗) |
| | $\Delta E_{PM-NM}$ (meV/f.u) | -935 | -1997 | -442 | -53 | -63 | -4313 | -1167 | -1925 | -130 |
| | $E_g$ (eV)/m ($\mu_B$) | 0.64 / 2.52 | 1.31 / 4.19 | 1.26 / 0.90 | 0 / 0.79 | 0 / 0.78 | 1.74 / 3.75 | 1.60 / 1.91 | 1: 0.06 / 3.94-3.07 2: 0 / 3.59 | 1: 0.50 / 1.26-0.16 2: 0 / 0.77 |

**Fig. 2** Summary of the properties (electronic configurations, space group symmetry, metal vs. insulator behavior, spin order, Goldschmidt tolerance factor, mode of gapping, energy differences between the different magnetic solutions, bandgap, and magnetic moments) of oxide perovskites with unpaired electronic $d$ shell configurations tested in our simulations. Green (red) tick marks refer to DFT success (failure) to reproduce the experimental observations available in the literature. For the high-temperature phase, the results of the nonmagnetic approximation to PM are provided in parenthesis

are required to properly describe them" in ref. [23], "spin polarized DFT shows metallic behaviour with neither magnetism nor bond disproportionation…: This qualitative structural error ….signals the importance of correlations" in ref. [17], or "While density functional band theory (DFT) is the workhorse of materials science, it does not capture the physics of the Mott/charge-transfer insulator transition" in ref. [19]. Moreover, gapping (and the related structural distortions) in the paramagnetic phases of oxides is often claimed to be unreachable by DFT simulations: "However, these methods (i.e., LDA, GGA, and LDA + U) usually fail to describe the correct electronic and structural properties of electronically correlated paramagnetic materials" and "Therefore, LDA + U cannot describe the properties of $LaMnO_3$ at $T > T_N$ and, in particular, at room temperature, where $LaMnO_3$ is a correlated paramagnetic insulator with a robust JT distortion" in ref. [18] or "Although they cannot represent the paramagnetic insulating state, static mean-field theories such as DFT, DFT + U, and hybrid functional approaches may capture some of the physics of the AFM insulating ground state" in ref. [17]. Many similar claims about the failure of DFT abound in the correlated literature.

However, naive DFT does not represent what DFT can do. There are certainly a number of papers over the years that have shown gapping in 3$d$ oxides using appropriate DFT and taking into account the structural, electronic, and magnetic degrees of freedom appearing in oxides (i.e., allowing spin polarization, lower energy phase searches for instance)[24–33]. With the exception of ref. [25], many of these studies focused on the low-temperature spin-ordered phase, even though the gapping usually appears in the paramagnetic (PM) spin-disordered phase. In ref. [25], Trimarchi et al. have proposed a strategy to model PM phases of simple binary oxides, identifying mechanisms to explain gapping in binaries such as MnO and NiO. This paper focuses on ternary compounds such as $ABO_3$ that possess

strongly entangled structural, electronic, and magnetic degrees of freedom, yielding far more complex physical behavior and a large range of functionalities encompassing ferroelectricity, magnetism, and thermoelectricity for instance. So far, DFT studies have neither addressed the PM phases of 3$d$ transition metal $ABO_3$ materials, nor elaborate the specific modalities of DFT required to produce gapping, nor have they systematically described the "Periodic Table of gapping" by considering the whole range of trends for the chemical series $ABO_3$ with different A and B atoms. The continuing impression is that DFT itself is failing. It thus seems that the question of what is the minimal theory that describes the basic ground-state phenomenology across the $ABO_3$ series—symmetry broken or symmetry conserving, statically (mean-field treatment of electron–electron repulsion as in DFT) or dynamically correlated—is still unsettled.

Following the standard E. Wigner definition[34], correlations are considered to be all physical effects beyond mean-field Hartree–Fock methods[35]. An often voiced popular (but incomplete) analogous view applied to DFT is that correlation is everything that DFT does not get right. According to this paradigm, the success of (a more general) DFT in describing the trends in the properties of $ABO_3$ $d$-electron perovskites, described in this paper, would suggest that whereas $ABO_3$ oxides may be complicated, they are not necessarily strongly correlated. In fact, DFT is an exact formal theory for the ground-state properties for the exact exchange-correlation energy functional[36], so there is no reason in principle why the properties noted above could not be captured by an ultimate DFT. The paradigm that correlation is what DFT does not get right is, therefore, most likely, a diminishing domain.

In this paper, we show that minimizing the total energy of the AFM and PM phases in supercell representations that allow symmetry breaking, produces atomic displacements and other symmetry-lowering modes that closely follow experiments, and

that the same symmetry-breaking modes also predict the correct trends in bandgaps and moments. This is achieved in a single-determinant, mean-field-like band structure approach. This suggests that dynamic correlations outside current DFT implementations and the view that symmetry-conserving physics applies across the board for all $AB(d)O_3$ $d$-electron systems are not forced upon us by the data, as suggested in previous literature, but represent instead a viewpoint of that literature. We find therefore that the ground-state properties of $3d$ $ABO_3$ oxides are not good examples of the failure of DFT or the need for special effects outside current DFT. Achieving this requires (a) allowing sufficient structural flexibility (a polymorphous description) in the description of the various phases, so that symmetry breaking reduced crystal symmetries that could lift degeneracies (octahedral rotations, Jahn–Teller, and bond disproportionation effects) and electronic instabilities could occur, should they lower the total energy, and (b) using an exchange-correlation functional in the Kohn-Sham DFT (KS-DFT) that distinguishes occupied from unoccupied states (such as DFT $+ U$; self-interaction corrected functionals) and minimizes the delocalization error[37]. We are neither claiming that all current and future $d$ or $f$-electron compounds can be fully explained by DFT with currently known exchange-correlation (XC) functionals, nor that properties other than the basic trends in gapping, moments, and structural displacements can be always predicted. These questions are left for future research. Such failures, if and when found, would provide legitimate challenges for explicitly correlated methodologies to explain. This is a consequential result because it suggests that a rather simple tool such as DFT (requiring a low computational effort with respect to heavier machineries treating dynamical correlations) offers a single platform to study reliably and with sufficient precision not only bandgap formation, structure, and magnetism in $ABO_3$, but also—in the future—other ground-state issues such as doping, defect physics, and interface effects.

## Results

**The DFT format used**. The dual input to this (or other) theories is: (a) a framework for interelectronic interaction (DFT exchange and correlation functional) and (b) a representation of the structural/geometrical degrees of freedom that could be optimized in seeking minimum total energy. Previously, technical shortcomings in (b) within a naive DFT approach (summarized in Supplementary Note 2) were often attributed to theoretical failures in describing the underlying interelectronic interactions (a).

The DFT features that are needed beyond the N-DFT are: (a) functionals that distinguish occupied from unoccupied states. Since we are interested in determining what are the minimum theoretical ingredients needed to explain the ground-state properties observed in the $ABO_3$ series, we deliberately adopt for (a) a mean-field, single-determinant, Bloch periodic band structure approach, with an electron-gas-based description of exchange and correlation. XC functional represents a hierarchy of approximations[38,39] ("rungs"). The first three rungs correspond to an XC potential depending on the local density (LDA) and its first (GGA) and second derivative (meta GGA), being local or semi-local functionals of the noninteracting density matrix, with no distinction between occupied and unoccupied levels. The first rung that distinguishes occupied from unoccupied states is rung 4 being a nonlocal functional of the noninteracting density matrix. While ideally this would be a fully self-interaction corrected functional, this functional does not exist as yet, so we use a simple approximation to it in the form of the DFT $+ U$ method.

A number of calculations have used this "DFT $+ U$" approach, where DFT is amended by an on-site potential that removes part

of the spurious self-interaction error and thereby creates a distinction between occupied and empty states producing at times gapped states[25,27–30,32,40–42]. DFT $+ U$ successfully obtained gapping in simple binary oxides such as MnO, FeO, CoO, or NiO[25], dioxides such as $UO_2$[26], and in the spin-ordered phases of the much more complex $3d$ transition metal $ABO_3$ compounds[27–32,41,43,44]. These successes have, in part, propagated the view that it is the interelectronic repulsion akin to the Hubbard Hamiltonian that produces gapping in DFT $+ U$. In fact, the role of the on-site potential $U$ in DFT $+ U$, where $U$ is a one-body on-site potential shifting the $d$ orbital to deeper energies, is distinct from the Hubbard Hamiltonian, where it truly represents interelectronic repulsion. Furthermore, $U$ in DFT is actually not required to produce gapping, as illustrated by $U$-free calculations for the spin-ordered state of several $ABO_3$ compounds[28,45–52] as well as for other complex oxides such as $VO_2$[24] and $La_2CuO_4$[53] for instance. Further details on the DFT calculations and on the choice of $U$ parameters are provided in Supplementary Note 3.

(b) The polymorphous representation of structural degrees of freedom: A primitive unit cell containing a single formula unit of $ABO_3$ cannot break the symmetry in band theory, as there is but a single $3d$ atom in the crystal. In order to provide a real test to the ability of the DFT XC functional to describe the basic trio of properties—spin arrangement, gapping, and structure—it is necessary to provide in (b) sufficient electronic and structural generality and flexibility in the way the system is represented, so that symmetry-breaking events requiring such flexibility could be captured insofar as they lower the total energy. Here, instead of restricting the unit cell representation of PM (and for AFM) magnetic structures to a single, primitive cell having but a single, symmetry-unique $BO_6$ motif (monomorphous representation), we allow supercells that can develop a set of symmetry-independent $BO_6$ motifs (polymorphous representation), subject to the constraint that the total supercell represents the macroscopic crystal and spin structures (e.g., zero moment in the PM). We use the Special Quasi Random (SQS)[54] construct that selects supercells of a given crystalline space group symmetry, such that the occupation of sites by spins follows a random pair and multibody correlation functions (appropriate to the HT limit) with a total moment of zero. The method is described in detail in refs. [25,54]. We have tested the convergence of the results with respect to energy, magnetic moment, and bandgaps on two "limit" compounds (namely $YTiO_3$, i.e., $d^1$, and $YNiO_3$, i.e., $d^7$). We have found that a 160-atom unit cell is sufficiently large to produce converged results (see Supplementary Note 4 for details on SQS and the generated supercells). This polymorphous representation[25] provides an opportunity to break spatial symmetry, should the total energy be lowered in doing so.

To explore the opportunities for symmetry breaking allowed, in principle, by the polymorphous approach, one has to exercise a few symmetry-breaking "nudges":

(1) Allowing different local environments for the various chemically identical $3d$ atoms in the lattice. Specifically, a different number of spin-up vs. spin-down sites can exist around each $3d$ site subject to the SQS constraint that all pair interactions are purely random (i.e., no short-range order) and the total spin is zero.

(2) Occupation number fluctuations whereby atomic sites with partial occupation of initially degenerate levels can have different assignments of the electrons to the degenerate partners [such as (1, 0, 1) for two electrons in the three partners of $t_{2g}$, rather than using fractional and equal occupation such as (2/3, 2/3, 2/3)][55]. The problem of finding the site occupations that lead to minimum energy is, in

general, a nontrivial optimization problem[26], especially when done for the artificial case of (i) a high- symmetry unit cell and (ii) a rigid lattice. However, here we (i) use a supercell that already has a low symmetry ($3d$ atoms have their own local environment) and (ii) allow the lattice to respond to site fluctuations in occupation numbers. In addition, as we will show below, most of the structural distortions ($O_6$ rotations and anti-polar motions of ions) appearing in the $ABO_3$ perovskites mix the "cubic" orbitals removing orbital degeneracies present in the initial cubic phase. Thus, the existence of metastable phases in the cubic cell is not so crucial for our study and these orbital broken symmetries are there to probe the mechanism yielding the bandgap (Jahn–Teller motion or disproportionation for instance). We have tested few initial guesses and only the most stable phase reached after the self-consistency is kept. Finally, for disproportionating materials (e.g., $YNiO_3$ or $CaFeO_3$), different types of initial nudging were performed, such as (1, 0) $e_g$ partner occupancies on all B sites or (1, 1) and (0, 0) $e_g$ partner occupancies between neighboring B sites.

(3) Displacement fluctuations (i.e., atomic relaxation requiring cells larger than the primitive cell), including local Jahn–Teller distortions as well as octahedral mode deformations, such as breathing, tilting, rotation, and anti-polar displacements described in detail in Supplementary Note 1.

**Predictions for low-temperature magnetically ordered AFM/FM phases**. Figure 2 summarizes our results for the lowest energy phase considering spin-ordered phases (energy differences between all tested magnetic configurations and initial symmetries are given in Supplementary Table 6). Consistent with experiments and previous DFT theoretical literature[27–30,32,40,41,56], for all explored compounds, we find (i) the correct low T crystal structure— orthorhombic for $CaVO_3$, $CaMnO_3$, $LaMnO_3$, and $LaFeO_3$; monoclinic for $LaVO_3$, $CaFeO_3$, and $YNiO_3$; cubic for $SrVO_3$; (ii) the correct low T spin-ordered phase, including AFM (for $YTiO_3$, $LaVO_3$, $CaMnO_3$, $LaMnO_3$, $LaFeO_3$, and $YNiO_3$) or FM (for $YTiO_3$) (except for $CaFeO_3$ that exhibits an incommensurate antiferromagnetic spin spiral order at low temperature[9] not included in our modeling); (iii) all compounds adopting a spin-ordered ground state ($YTiO_3$, $LaVO_3$, $CaMnO_3$, $LaMnO_3$, $CaFeO_3$, $LaFeO_3$, and $YNiO_3$) are predicted insulating. Furthermore, (iv) the key cell-internal lattice distortions ($O_6$ group rotations, Jahn–Teller distortions, and bond disproportionation) observed experimentally, are reproduced by

theory with mode amplitudes in excellent agreement with experiments (see Supplementary Table 7).

**Predictions for high-$T$ paramagnetic phases**. Predicted structure types and sublattice distortions: Using the polymorphous representation, the DFT + SQS reproduces the experimentally observed structure for PM phases ($YTiO_3$, $CaVO_3$, $SrVO_3$, $LaVO_3$, $CaMnO_3$, $LaMnO_3$, $CaFeO_3$, $LaFeO_3$, and $YNiO_3$) as summarized in Fig. 2. Interestingly, the relaxed HT structures share key similarities with the LT phases ($O_6$ rotations, bond disproportionation), as inferred by our symmetry-adapted mode analysis (Supplementary Table 7), i.e., the LT phases generally inherit the properties of the PM phases. Only $LaVO_3$ exhibits a PM phase that undergoes an alternative Jahn–Teller motion pattern with respect to the LT phase, yielding an orthorhombic symmetry instead of a monoclinic symmetry (Supplementary Table 7 and Supplementary Fig. 1).

Predicted metal versus insulating characteristics: The present approach correctly reproduces the experimentally observed insulating or metallic behaviors for all materials (Fig. 2). We note, however, that just as is the case in the highly uncorrelated compounds Si and GaAs, DFT often does not give accurate absolute bandgaps (and hence, effective masses), and a GW correction is needed[57]. The $3d$ oxides of $ABO_3$ are no exception; we expect that more quantitative gap values and (renormalized) masses will be available as GW is applied to our SQS–DFT.

Surprisingly, the approach also correctly reproduces the two experimentally observed paramagnetic phases of $CaFeO_3$ and $YNiO_3$, with an insulating phase slightly more stable than the metallic phase (Fig. 2; Supplementary Table 6). Finally, we note that $ABO_3$ with early $3d$ elements B = Ti, V shows band edges that are $d$-like (upper Hubbard and lower Hubbard bands (c.f. Fig. 1b, top), whereas those with later $3d$ elements (B = Mn, Fe, and Ni) display oxygen-like band edges (see Figs. 3–6).

**Gapping mechanisms**. Having now established that the methodology used correctly captures the basic physical properties observed experimentally for the different compounds presented in Fig. 2, we next study the triggering mechanism of the gap formation, i.e., the factors causing gapping, providing a classification of gapping mechanisms for the different $ABO_3$ perovskites.

The methodology used to identify the leading gapping mechanism is to start from an assumed high-symmetry cubic perovskite phase (Pm-3m symmetry), and slightly "nudge"

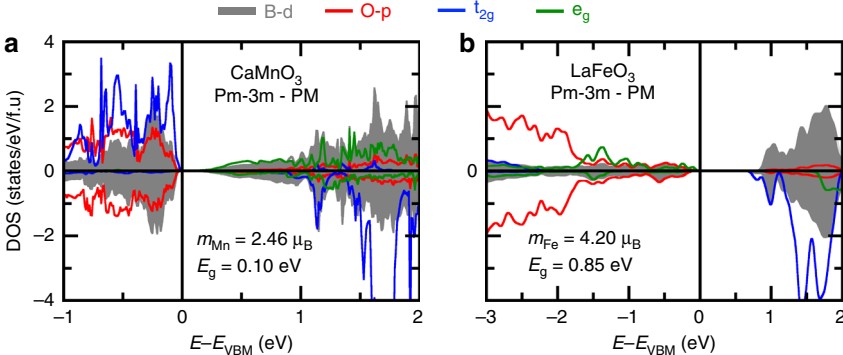

**Fig. 3** Electronic properties of compounds with closed subshells. Averaged projected density of states on B $d$ levels (gray) and O $p$ levels (red) in $CaMnO_3$ (**a**) and $LaFeO_3$ (**b**) in a hypothetical cubic phase within the PM order. The averaged density of states is extracted by summing all contributions coming from all atoms in the PM cells, in each of the spin channels, for $p$ (O, in red) and $d$ (B, in gray) orbitals. Positive (negative) values stand for spin up (down). Orbital resolved DOS on $t_{2g}$ (blue) and $e_g$ (green) level is also provided, but for a given B-site cation in the supercell

(in the linear response sense) this structure with respect to potential symmetry- breaking modes, looking for energy lowering and gap formation. The perturbing modes are (a) allowing different local environments, (b) occupation number fluctuation, and (c) displacement fluctuations, including local Jahn–Teller distortions as well as octahedral mode deformations, such as breathing, tilting, rotation, and anti-polar displacements. Once the leading symmetry- breaking mechanism is found (with respect to cubic), we proceed to perform the complete gapping and magnetism calculation using the actual crystal structure (orthorhombic, cubic, and monoclinic) looking for minimum energy. We group the $ABO_3$ compounds into four categories in terms of the gapping mechanisms.

### Gapping mechanism 1: compounds forming closed subshells by octahedral breaking of atomic symmetry.

The free ions $Mn^{4+}$ ($d^3$) and $Fe^{3+}$ ($d^5$) with an odd number of electrons would simplistically lead to partially filled degenerate states and hence candidate for ungapped "metallic bands". But the solid-state octahedral symmetry breaks the continuous rotational atomic $d$ symmetry: starting from an assumed cubic phase, $CaMnO_3$ and $LaFeO_3$ are already found to be insulators with bandgaps of 0.10 eV ($CaMnO_3$, $t_{2g}^3$) and 0.85 eV ($LaFeO_3$, $t_{2g}^3 e_g^2$) and sizable magnetic moments of 2.46 $\mu_B$ and 4.20 $\mu_B$, respectively (Fig. 3a, b). Consequently, in this group, the bandgap originates from lowering the atomic symmetry by the octahedral field and Hund's rule, driving half-filling of the degenerate partners and an ensuing gap already at the cubic level (such an observation was already raised for $CaMnO_3$[31]). Because of the relative A-to-B atomic size mismatch in $ABO_3$ (as reflected by the 1926 Goldschmidt tolerance factor $t$[58] reported in Fig. 2, being less than 1), these compounds do not prefer the ideal cubic structure and could distort by $O_6$ rotations, as indeed obtained by DFT energy minimization. This additional symmetry lowering produces energy gains of 315 and 378 meV/f.u in $CaMnO_3$ and $LaFeO_3$, respectively, further increasing their bandgaps to 0.64 and 1.31 eV without altering the B cation magnetic moment (Fig. 2).

### Gapping mechanism 2: compounds lifting electronic degeneracies due to octahedral rotations and those retaining electronic degeneracies due to small or absent octahedral rotations.

Unlike $CaMnO_3$ and $LaFeO_3$, the cubic symmetry alone does not create half-filling in $CaVO_3$, $SrVO_3$, $YTiO_3$ ($d^1$), and $LaMnO_3$ ($d^4$) that continues to exhibit electronic degeneracies of either $t_{2g}$ or $e_g$ levels. $CaVO_3$ and $SrVO_3$ are metals, whereas $YTiO_3$ and $LaMnO_3$ are insulators in both PM and LT phases. These can be classified in terms of the strength of the octahedral tilting effects.

$YTiO_3$ and $LaMnO_3$ are gapped insulators due to octahedral rotation in the absence of electronic instability: We first probed the $d^1$ orbital degeneracy of $Ti^{3+}$ in $YTiO_3$ and of $d^4$ of $Mn^{3+}$ in $LaMnO_3$ by nudging the occupancy to a specific degenerate partner within the cubic phase (e.g., occupation of the $t_{2g}$ triplet by (1, 0, 0) rather than (1/3, 1/3, 1/3)). We find that the imposed orbital broken symmetry (OBS) is unstable and decays back to a metallic solution with equal occupations of degenerate partners. Thus, in the ideal cubic symmetry, these compounds do not develop any electronic instability that would break the $d$ orbital degeneracy, thereby leading to gap formation.

Permitting the next non-cubic symmetry breaking reveals the gapping mechanism here. Due to a large A-to-B cation size mismatch in $ABO_3$, reflected by their tolerance factor (Fig. 2), $YTiO_3$ and $LaMnO_3$ are expected to be unstable in their cubic structures and to develop $O_6$ rotations, lowering the symmetry from cubic Pm-3m to orthorhombic Pbnm, as indeed found in DFT energy minimization. This results in large total energy lowering ($\Delta E = -1898$ and $-632$ meV/f.u in $YTiO_3$ and $LaMnO_3$, respectively) and in insulating phases with a bandgap created between $d$ levels (Fig. 4a, c). Due to the symmetry lowering, the point group symmetry is reduced from $O_h$ to $D_{2h}$ and a new basis of $d$ orbitals is produced locally on each of the transition metal sites. In $YTiO_3$, it results in a split-off $d$ band below the Fermi level (Fig. 4a) and the Ti $d$ electron is localized in an orbital corresponding to a linear combination of the "cubic-$t_{2g}$" levels (Fig. 4b) of the form $\alpha d_{xy} + \beta d_{xz} + \gamma d_{yz}$ ($\alpha^2 + \beta^2 + \gamma^2 = 1$), whose coefficients $\alpha$, $\beta$, and $\gamma$ are triggered by

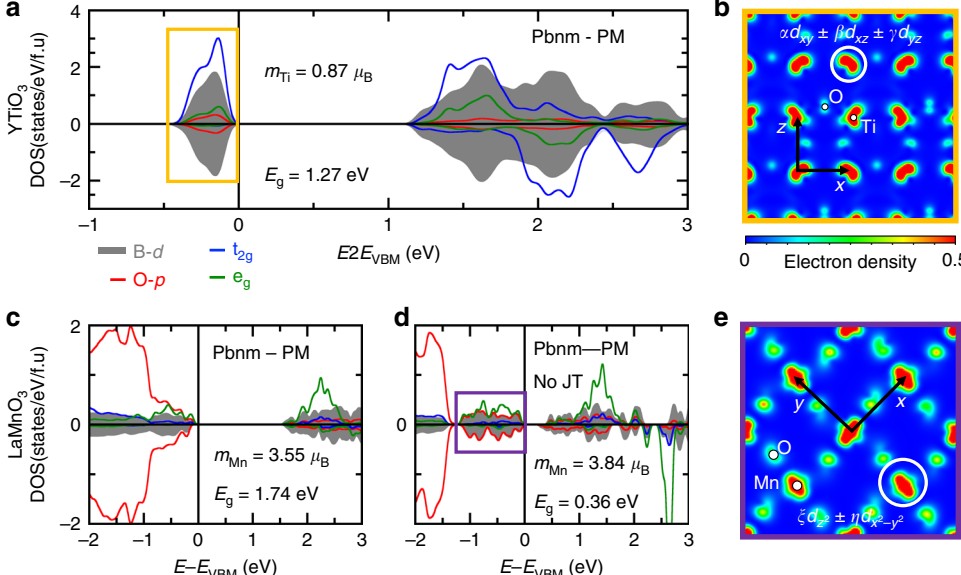

**Fig. 4** Electronic properties of compounds lifting electronic degeneracies through octahedra rotations. **a, c, d** Averaged projected density of states on B $d$ levels (gray) and O $p$ levels (red) in $YTiO_3$ (B = Ti, upper panel) and $LaMnO_3$ (B = Mn, lower panel) in the PM phase. Projected density of states on $t_{2g}$ (blue) and $e_g$ (green) levels for a specific B cation within the supercell are also reported. **b, e** Partial charge-density maps in the (*ab*) plane of levels located near the Fermi level reported in **a** and **d**

combinations of octahedra rotations and A cations anti-polar motions[27] (no Jahn–Teller (JT) motions are observed in the structure on the basis of our symmetry-adapted modes presented in Supplementary Table 7). This results in "orbital ordering" that is clearly a DFT band structure effect triggered by lowering the site symmetry, not a dynamical correlation effect. The characteristic shape of the orbital (reflected in the charge-density map of Fig. 4b) then drives the FM interactions at low temperature[27].

A similar mechanism applies for LaMnO₃, although the material displays a large in-phase JT distortion (Supplementary Table 7). This produces a gap located between $e_g$ levels (Fig. 4c). To settle the importance of the additional JT motion with respect to YTiO₃, we calculate the very same Pbnm phase but remove the JT mode. We observe that the material is already an insulator and the averaged $Mn^{3+}$ magnetic moments are mostly unaltered (Fig. 4d) (even though relying on standard LDA calculations, rotations were previously shown to open pseudo gaps, irrespective of the presence of a Jahn–Teller motion in ref. [59]). Therefore, $O_6$ rotations are sufficiently large to split the new basis of $e_g$ levels and the $Mn^{3+}$ electrons are localized in an orbital pointing either along $x$ or $y$ directions on Mn neighboring sites in the (ab) plane (Fig. 4e). These orbitals correspond to linear combinations of the "cubic $e_g$" levels of the form $\xi d_{z^2} + \eta d_{x^2-y^2}$ $(\xi^2+\eta^2=1, |\xi| \approx |\eta|)$. The specific orbital pattern stabilizes an AFM-A order at low temperature, without changing substantially the bandgap and the averaged magnetic moments (Fig. 2).

CaVO₃ and SrVO₃ are metals because of insufficiently large $O_6$ rotations: because of their closer-to-1 tolerance factor (Fig. 2), CaVO₃ and SrVO₃ are not developing large $O_6$ rotations (Supplementary Table 7), as found in DFT energy minimization, and thus the orbitals are not split to render an insulating phase. Note that the metallic state is not due to some strongly correlated effect but rather due to a trivial, semiclassical mechanism captured by the 1926 Goldschmidt tolerance factor[58]. The mechanism was validated by artificially imposing the YTiO₃-type $O_6$ rotations onto CaVO₃, finding in our variational self-consistency, an insulator with a bandgap of 0.14 eV. In agreement with this result, Pavarini et al. noted in their dynamical mean-field theory (DMFT) simulations[21,60,61] on YTiO₃ and LaMnO₃ that orthorhombic distortions (rotations, A-cation motions) can produce a sufficiently large crystal field splitting localizing electrons, irrespective of Jahn–Teller distortions.

It is often thought that SrVO₃ is a good example of failure of mean-field-like theories, and that SrVO₃ shows a correlated behavior beyond Hartree–Fock descriptions, i.e., formation of Hubbard bands and a coherent peak at the Fermi level $E_F$ due to

the electronic correlation in the ground state, revealed by photoemission experiments. This viewpoint classifies SrVO₃ into a strongly correlated metal category and dynamical correction in the ground-state description is surely necessary to describe the basic property of the correlated perovskites. We emphasize that the metallic nature of SrVO₃ is a simple consequence of the cubic structure being the lowest total energy phase. We note that the SQS-PM approach that is deliberately restricted to a single-determinant mean field gives a clear upper Hubbard band and this bandwidth $W$ associated with $t_{2g}$ is 1.6 eV (SQS-PM DFT), matching well the angle-resolved photoemission spectroscopy[62] (ARPES) that gives $W_{t_{2g}} = 1.3$ eV and DMFT[63–65] in which $W_{t_{2g}}$ is estimated to 0.9, 1.2, or 1.35 eV depending on the DMFT parameters. In contrast, N-DFT using a single monomorphous unit cell gives a bandwidth $W_{t_{2g}}$ of 2.6 eV as cited by ref. [65].

**Gapping mechanism 3: compounds with two electrons in $t_{2g}$ levels.** Although a metallic behavior would be expected for LaVO₃ ($t_{2g}^2 e_g^0$) with its two electrons distributed in three $t_{2g}$ partners, it is experimentally found to be insulating at all temperatures and no metal–insulator phase transition has been reported in its bulk form[6].

In order to understand the mechanism for gapping in this group, we initially break orbital occupancies by nudging two electrons in two out of three $t_{2g}$ orbitals on the $V^{3+}$ site in the assumed high-symmetry cubic phase (e.g., (1, 1, 0) $t_{2g}$ occupancies). Following variational self-consistency, LaVO₃ reaches a lower total energy ($\Delta E = -321$ meV/f.u with respect to the (2/3, 2/3, 2/3) degenerate cubic phase) with an insulating electronic structure having a bandgap located between two occupied and one unoccupied formerly $t_{2g}$ levels (Fig. 5a). From this imposed orbital broken symmetry, we conclude that the electronic structure is unstable in the high- symmetry cubic phase of rare-earth vanadates and that these materials should therefore have a secondary contribution to the bandgap due to a Jahn–Teller distortion.

This can be verified by allowing LaVO₃ to lower its energy within the PM spin order, by letting it develop $O_6$ rotations and anti-polar motions, resulting in an orthorhombic symmetry. This structure develops a sizable in-phase Jahn–Teller motion similar to that displayed by LaMnO₃, albeit smaller (Supplementary Table 7), and is insulating with a large bandgap of 1.60 eV located between $t_{2g}$ levels (Fig. 5c). Again, the JT distortion is but secondary for the gap opening, as evidenced by the fact that a Pbnm distorted structure in which we artificially eliminated the JT distortion already exhibits similar characteristics in terms of magnetic moments and bandgap value (Fig. 5b). Rotations driven by pure steric effects, showing energy gain around 700 meV/f.u

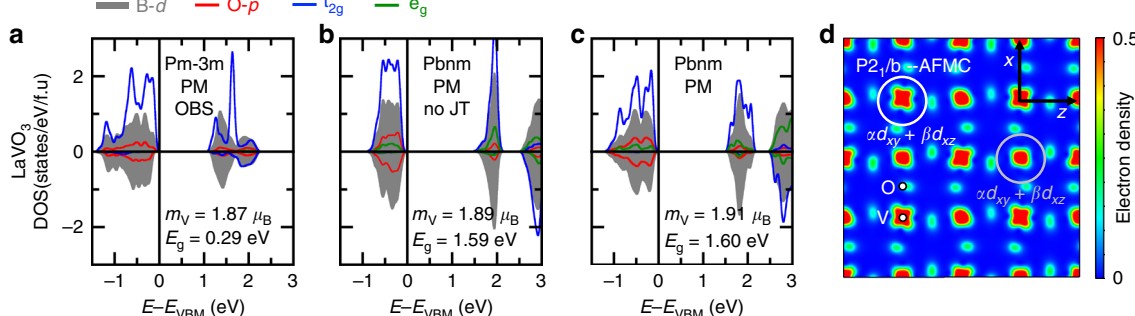

**Fig. 5** Electronic properties of compounds with an unstable electronic structure. **a–c** Averaged projected density of states on V $d$ levels (gray) and O $p$ levels (red) in LaVO₃ in the PM phase with different symmetries, lattice distortions, or orbital broken symmetries (OBS). Projected density of states on $t_{2g}$ (blue) and $e_g$ (green) levels for a specific V cation within the supercell are also reported. **d** Partial charge-density plot in the (xz) plane of the last two occupied bands in the AFM-C phase of LaVO₃

with respect to the degenerate cubic phase (compared with the 321 meV/f.u produced by the intrinsic electronic instability), are again sufficient to alleviate orbital degeneracies.

At low temperatures, $LaVO_3$ transforms to an AFM-C insulator. This LT phase develops an alternative and sizable Jahn–Teller motion (Supplementary Table 7), for the octahedral distortion is in antiphase along the c axis, while it is in-phase for the JT motion appearing in Pbnm phases (Supplementary Fig. 1). This alternative JT motion lowers the symmetry from Pbnm to $P2_1/b$ and produces a specific orbital pattern, different from that appearing in Pbnm phases (Fig. 5d; Supplementary Fig. 3). The extra d electron with respect to $YTiO_3$ is localized in a rock-salt pattern of an orbital corresponding to either a $ad_{xy} + \beta d_{xz}$ or a $ad_{xy} + \beta d_{yz}$ ($\alpha^2 + \beta^2 = 1$) combination. The orbital pattern favors an AFM-C order through the Kugel–Khomskii mechanism[66] without significantly changing the bandgap and magnetic moments (Fig. 2).

We see that due to pure steric effects, the $O_6$ rotations are sufficiently large to produce a new basis of orbitals, rendering a localized state in the Pbnm symmetry. The JT motion appearing in the LT phase is reminiscent of the native electronic instability of these materials[29] and of the "orbital-order" phase transition reported experimentally[6]. Finally, the observation of the insulating phase in the orthorhombic symmetry without Jahn–Teller motion is again compatible with DMFT simulations of Pavarini et al. on $YVO_3$ and $LaVO_3$[20].

## Gapping mechanism 4: compounds with unstable single local electronic occupation patterns disproportionating into a double local environment.

Although $CaFeO_3$ and $YNiO_3$ develop an electronic degeneracy similar to that of $LaMnO_3$ with a single $e_g$ electron, we note that $CaFeO_3$ and $YNiO_3$ are metallic within the orthorhombic Pbnm PM phase, and become insulating in a lower symmetry space group—the $P2_1/n$ monoclinic structure. We present here details for $YNiO_3$; the very same conclusions are drawn for $CaFeO_3$ in Supplementary Note 8.

We start from an ideal Pm-3m cubic phase and artificially offer breaking of the degeneracy of the $Ni^{3+}$ $t_{2g}^6 e_g^1$ levels by forcing a specific $e_g$ partner occupancy (e.g., (1, 0) instead of (0.5, 0.5)). However, the imposed orbital broken symmetry does not survive the variational self-consistency and the $e_g$ electron spreads equally on the two orbitals. We offer an additional symmetry-breaking route by breaking of the degeneracy of the $Ni^{3+}$ levels via forcing $e_g$-level occupancies on two different Ni sites to be (1, 1) and (0, 0) respectively. This yields a rock-salt pattern of Ni sites with half-filled and empty $e_g$ levels, respectively, i.e., mimicking Ni sites with 2 + and 4 + formal oxidation states (FOS). Following variational self-consistency, $YNiO_3$ is trapped in such a state that proves to be of lower energy by 15 meV/f.u compared with the degenerate cubic phase, still with degenerate $e_g$ levels on each Ni site and no gap (Fig. 6a). In this OBS, we detect slightly different electronic structures between neighboring transition metal sites (labeled $Ni_L$ and $Ni_S$), yielding different $e_g$-level occupancies and magnetic moments (Fig. 6c). We determine from this probing of the linear response of the system to orbital nudging that the electronic structure is latently unstable and prone therefore to yield distortions in order to produce two types of B environments, one with half-filled ($Ni^{2+}$) and one with empty ($Ni^{4+}$) $e_g$ states.

Building on our determination of the role of electronic instability, we can now perform the structural relaxation starting from the cubic phase with or without the OBS. Without OBS, $YNiO_3$ relaxes to an orthorhombic Pbnm symmetry ($\Delta E = -921$ meV/f.u energy lowering with respect to the cubic phase without OBS) that is metallic. This phase is characterized

by the usual $O_6$ rotations and by a single local environment (SLE) for B cations. With OBS, we find that the material relaxes to a $P2_1/n$ monoclinic phase that is insulating with a gap of 0.50 eV (Fig. 6b) and more stable than the Pbnm symmetry ($\Delta E = -23$ meV/f.u with respect to the Pbnm phase, Fig. 6f). Along with the usual Pbnm structural distortions, this phase develops a striking feature: there is a bond disproportionation of two $O_6$ groups, producing a rock-salt pattern of collapsed and extended octahedral (for which Ni cations sitting at their center are labeled $Ni_S$ and $Ni_L$, respectively) due to a sizable breathing mode (Supplementary Table 7), resulting in a double local environment (DLE) for B cations, each having very different B–O bonds. Consistently with the electronic instability identified in the cubic cell, the DLE results in different electronic structures for $Ni_L$ and $Ni_S$ ions, characterized by two magnetic moments, the former larger than 1 and the latter approaching 0 (Fig. 6b, d). This is also reflected by the partial charge-density maps (Fig. 6e) obtained from electronic states located between −1 eV and the Fermi level, showing that electrons prefer to occupy $e_g$ levels on $Ni_L$ sites. Thus, $YNiO_3$ reaches insulation through disproportionation effects transforming the 3 + unstable formal oxidation state (FOS) of Ni ions, to its more stable 2 + /4 + stable FOS in the insulating phase[67].

Remarkably, although the observation of 2 + /4 + FOS in the insulating phase is consistent with the Wannier analysis discussed in ref. [28], the physical charge density $\rho(\vec{r})$ around $Ni_S$ and $Ni_L$ cations is nearly indistinguishable—and similar to that extracted in the metallic Pbnm or cubic phases—as clearly seen in Fig. 3 in ref. [68] owing to a charge self-regulation mechanism[68–70], whereby the ligands resupply the 3d site with charge lost. This result has sometimes been interpreted in terms of a "ligand-hole" language in refs. [17,28,71] Thus, there is no apparent "physical charge ordering", in agreement with previous DMFT[17] or DFT + $U$[28] studies. That the extent of charge transfer, the lack of physical charge ordering, and the magnitude of disproportionation are reasonably depicted by DFT can be judged by examining the ensuing calculated bond lengths vs. experiment: Supplementary Table 7 shows that the predicted amplitude associated with the bond disproportionation distortion is in excellent agreement with experimental quantities in $CaFeO_3$ and $YNiO_3$.

We can finally check the relative role of the amplitude associated with rotations with respect to the disproportionation effects (we recall that, alone, the latter does not open a gap in $YNiO_3$). To that end, we have performed calculations for other $RNiO_3$ members (R = Gd, Sm, Nd, and Pr) in which $O_6$ rotations progressively decrease (Supplementary Note 9). Surprisingly, we find that only R = Gd and R = Sm are relaxing to a monoclinic phase with a DLE, while R = Nd and Pr are more stable within the orthorhombic symmetry and a SLE (Fig. 6f). Moreover, only R = Y, Sm, and Gd become insulating, while the other two compounds remain metallic (Fig. 6g). We conclude that the $O_6$ rotation amplitudes are controlling the ability to disproportionate to an insulating PM phase in nickelates.

However, we find that all the considered nickelates exhibit an AFM-S order, based on ↑↑↓↓ spin chains of Ni cations in the (ab) plane with different stackings along the c axis[72], in their ground state. It thus produces a monoclinic $P2_1/n$ symmetry that is insulating. We observe that the AFM-S phase shares all the key features of the monoclinic $P2_1/n$ PM phase, except the fact that the magnetic moment associated with $Ni_S$ cations becomes exactly null. The complex AFM-S order, which is compatible with the symmetry of the breathing mode, is therefore crucial to open the bandgap via forcing disproportionation effects when materials develop small $O_6$ rotations.

Compounds with an unstable electronic configuration in a SLE structure therefore reach insulation through several sequential

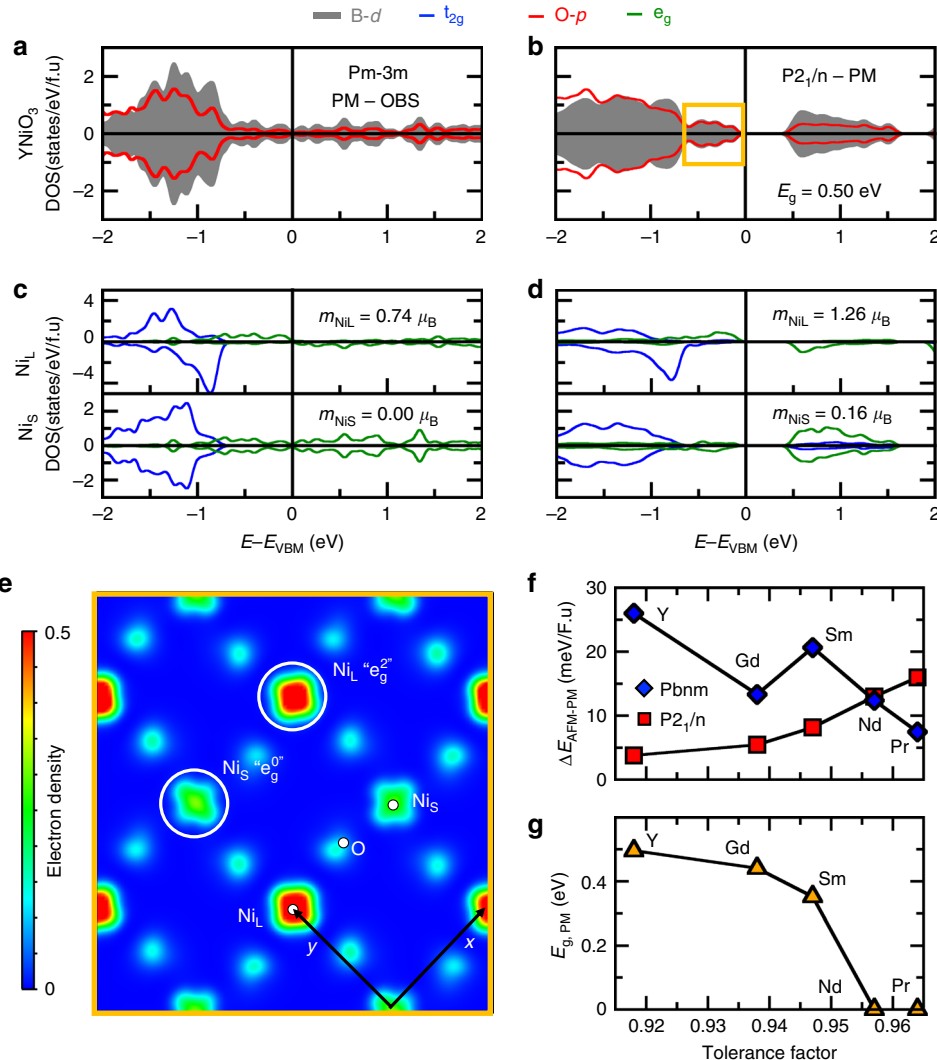

**Fig. 6** Electronic properties of compounds undergoing disproportionation effects. **a**, **b** Averaged projected density of states on Ni $d$ levels (gray) and O $p$ levels (red) in YNiO$_3$ in the cubic (**a**) and monoclinic (**b**) PM phase. **c**, **d** Projected density of states on t$_{2g}$ (blue) and e$_g$ (green) levels for a couple of Ni$_L$ (upper panel) and Ni$_S$ (lower panel) cations within the supercells. **e** Partial charge-density maps in the (**a**, **b**) plane of levels located near the Fermi level (the energy window is reported in **b**). **f** Energy difference (in meV/f.u) between the AFM ground state and PM solutions using the orthorhombic (blue diamond) and monoclinic (red squares) symmetries as a function of the tolerance factor. **g** Bandgap $E_g$ (in eV) associated with the lowest PM phase as a function of the tolerance factor

factors: (i) they possess an electronic instability, yielding disproportionation effects already in the high-symmetry cubic phase to get rid of the B cation unstable electronic configuration originating from an unstable formal oxidation state; (ii) the disproportionation is strongly linked to the amplitude of the octahedral rotations, appearing first due to steric effects; (iii) antiferromagnetic interactions force the disproportionation when O$_6$ rotations are weak. Our results reconcile the experimental phase diagram of rare-earth nickelates, and most notably, the PM metal to PM insulator or PM metal to AFM-insulator as a function of the rare-earth ionic radius[12]. Moreover, we unify the different models proposed to explain the metal–insulator phase transition (MIT) with (i) the existence of an electronic instability in rare-earth nickelates (DMFT[73,74] and DFT with AFM order[30]) and (ii) a structurally triggered Peierls MIT in RNiO$_3$ and AFeO$_3$ (A = Ca, Sr) compounds, although the DFT calculations were restricted to simple spin-ordered states[30,40].

Finally, the disproportionation effect was described in terms of a "novel correlation effect"[17], but as seen here, and in refs. [28,68],

it is predictable by the static mean-field DFT. The very recent DMFT simulations of RNiO$_3$ compounds performed by Hampel et al.[74] closely match our DFT + $U$ results presented in Fig. 6f, g.

## Discussion

Our DFT calculations of both LT magnetically ordered and HT spin-disordered PM phases reveal the minimal theoretical ingredients required to explain the trends in metal–nonmetal behavior in oxide perovskites, and the associated trends in forms of magnetism and structural selectivity. This includes spin polarization, magnetic interactions, and lower energy phase search through the polymorphous representation—allowing large enough (super) cells, so that various modalities of structural and electronic symmetry breaking can exercise their ability to lower the total energy. This leads one to identify four generic mechanisms opening a bandgap in oxide perovskites. Two mechanisms are related to purely structural symmetry breaking, such as natural octahedral symmetry and its associated

symmetry-lowering distortions, such as octahedral tilting and rotations, due to steric effects. The two other mechanisms are related to intrinsic electronic instabilities of transition metals, originating either from spontaneous orbital symmetry breaking, i.e., Jahn–Teller effect or an unstable formal oxidation state of transition metals (disproportionation effect), both manifested by dedicated structural distortions.

Previous statements that DFT fails in describing gapping trends in Mott insulators were premature and often based on a naive application of DFT without properly exploring channels of energy-lowering symmetry breaking, rather than on the failure of the description of interelectronic interactions underlying the density functional theory itself. Indeed, the sequential steps of offering to the system symmetry-breaking modes appear to explain gapping, the nature of the magnetic order, and both the space group symmetries and sublattice distortions (Jahn–Teller and octahedral rotations) in both low temperature and high temperature phases of all $ABO_3$ perovskites with B = 3d. This approach does not require explicit dynamical correlations, or the Mott picture of electron localization on atomic sites with the ensuing formation of upper and lower Hubbard bands (Fig. 1). The basic interactions of crystal field splitting, lattice distortions, and spin polarizations are sufficient to produce insulation in these compounds, and the celebrated Mott–Hubbard mechanism for gap opening (Fig. 1a) may not apply so generally to perovskite oxides. Thus, although these oxides are certainly complicated, they are not obviously strongly dynamically correlated materials and are not good examples where DFT with current exchange-correlation functionals fails.

## Data availability

All data are available upon reasonable request to the corresponding author.

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

## Acknowledgements

This work of J.V. has been supported by the European Research Council (ERC) Consolidator grant MINT under contract no. 615759. Calculations took advantages of the Occigen machines through the DARI project EPOC A0020910084 and of the DECI resource FIONN in Ireland at ICHEC through the PRACE project FiPSCO. The work of A.Z. was supported by the Department of Energy, Office of Science, Basic Energy Science, MSE division under grant no. DE-FG02–13ER46959 to CU Boulder. We acknowledge discussions with G. Trimarchi and Zhi Wang and technical support from A. Ralph at ICHEC supercomputers.

## Author contributions

J.V. did all calculations and analysis and participated in writing of the paper. A.Z. suggested the problem, did the analysis of the results, and directed the writing of the text with contributions from J.V. and M.B. M.B. provided experimental feedback and assisted in the writing of the paper.

## Additional information

**Competing interests:** The authors declare no competing interests.

