## [Peer Review File · Nature Communications]

Reviewers' comments:

Reviewer #1 (Remarks to the Author):

i read the present paper with great interest. If i can summarize, i would say the key point the paper points out is that KS-DFT (Kohn-Sham DFT) is often wrongly stated to "fail" to describe e.g, the band gaps of transition metal perovskites. Such "failure" is then taken as evidence of strong correlation effects. In fact, as the authors point out, such "failures" are usually the result of the said community either ignoring relevant and ubiquitous degrees of freedom such as structural distortions or not understanding the relationship between what they are modeling and physical reality such as pretending a non-magnetic KS-DFT calculation (zero local moment) has any physical relevance to a transition metal oxide in the paramagnetic phase (where local moments have formed at much much higher temperatures than at which they order). As you might have inferred from the above, i agree completely with the conclusions of the authors. I think this is an important discussion to have, and am pleased the authors have taken the time to write such a paper. that being said, it isn't clear to me that the "tone" of the paper is appropriate and/or the claims taken at face-value are necessarily correct.

1. First, how is the key point of this paper different from a previous of one of the authors?

Polymorphous band structure model of gapping in the antiferromagnetic and paramagnetic phases of the Mott insulators MnO, FeO, CoO, and NiO
Giancarlo Trimarchi, Zhi Wang, and Alex Zunger
Phys. Rev. B 97, 035107 – Published 5 January 2018

2. i have two competing views of this paper: 1) this is an important paper that needs to be published in a journal such as Nat Comm, 2) this is an artificial problem in the sense that anyone who works on transition metal oxides, and knows what they are doing, already understands these issues. with regard to the latter, i think the authors need to do a much better job of citing the existing literature where researchers have pointed out similar conclusions as the authors here. I would like to emphasize that i think the present paper is still important, as nothing published was done as systematically nor was performed across the perovskite "period table."

3. The present paper focuses a lot on the "plus U" method in KS-DFT and how people state that since ks-dft fails one needs to use the plus U method to treat correlations. They correctly point out that it does not capture the physics of the Hubbard model. In my mind, the plus U method is best thought of as a poor-mans self-interaction correction, and the authors even say as much. so the question/comment:

3a. a central aspect of the authors argument is that since "DFT" and subsequently "plus U" is a single-determinate theory, it doesn't have correlation effects. BY this logic however, is it true that any model-Hamilton theory solved within mean field theory doesnt have correlations? I do understand that the proper definition of a "Correlated" system is one in which cant be described by a single determinate, however, i think the entire many-body community (for lack of a better way to describe them) would still refer to the system as correlated even if a mean field theory describes the physics. I think this is a language issue, not physics?

3b. I know the authors know this, but you consistently refer to DFT as a single determinate theory. Dont you mean Kohn-sham DFT? Also, please take a step back and reread your paper, there are several places where seem to suggest that DFT is not an exact theory of the many-body state. which brings me to a question or comment. I have to confess that i am not an expert on the theoretical structure of DFT itself, but my understanding is that even Kohn-Sham DFT is an exact many-body theory, we just don't know the Exchange Correlation functional. So then isnt it natural that a "better" EC-functional, contains a "better" description of correlations? So wrt the discussion

of the meta-gga and band gaps, it isn't clear to me the meta-gga works "better" because of a "better" description of correlations. In fact, I think this is what the authors are saying indirectly? I have to say, it seems this paper was written to "preach" to one community, the community who loudly mistakenly claim "DFT" does not describe correlations (this is what I meant by "tone" in my first paragraph). In doing so, those who actually know better can misread some of the statements in the paper. (I again want to state, I know the authors are in the camp who "know better", but I think they would benefit if they "take a step back" and think about who will read this paper.)

Reviewer #2 (Remarks to the Author):

The authors systematically studied the magnetic, electronic, and structural properties of a set of prototypical ABO₃ perovskites, and found that within DFT augmented by the empirical U to correct the intrinsic self-interaction error all the properties can be predicted to be consistent with experimental results, including the paramagnetic phases under the SQS model, via symmetry-breaking energy-lowering procedures. In particular, all insulators were predicted to have gaps. To eliminate the complication caused by U, they applied the SCAN density functional to those insulating phases, and all predictions for insulators remained valid. The authors therefore concluded that whereas ABO₃ oxides may be complicated, they are not necessarily strongly correlated. As these compounds are typically regarded as strongly correlated materials, where strong inter-electron interaction is thought to be critical and computationally efficient conventional density functionals not applicable, this study will have a significant impact on studies of traditionally defined strongly correlated materials. The results the authors presented are solid and the analysis of gapping mechanism should be interesting to a wide audience. I therefore recommend the manuscript for publication in Nat. Comm.

The following are several suggestions:

- 1) The authors should discuss how well the SQS model models the PM phases since this is one of the critical points in this manuscript.
- 2) In the occupation number fluctuations, the method to lift the degeneracy by nudging the occupation, the authors should discuss about the metastable states encountered in DFT+U, as discussed in B. Meredig, et al, PRB, 82, 195128 (2010). This problem makes DFT+U less ideal for this study, which however has been alleviated by the application of SCAN on insulators.
- 3) For this work, when citing SCAN, it is appropriate to also cite J. Furness, et al, Communication Physics, 1, 11 (2018), where SCAN is demonstrated to open the gap for the pristine La₂CuO₄.
- 4) At page 6, use (1), (2), and (3) instead of (a), (b), and (c) for the symmetry breaking channels.

Author reply to Comments of Reviewer #1

1. Referee: I read the present paper with great interest. If I can summarize, I would say the key point the paper points out is that KS-DFT (Kohn-Sham DFT) is often wrongly stated to "fail" to describe e.g. the band gaps of transition metal perovskites. Such "failure" is then taken as evidence of strong correlation effects. In fact, as the authors point out, such "failures" are usually the result of the said community either ignoring relevant and ubiquitous degrees of freedom such as structural distortions or not understanding the relationship between what they are modelling and physical reality such as pretending a non-magnetic KS-DFT calculation (zero local moment) has any physical relevance to a transition metal oxide in the paramagnetic phase (where local moments have formed at much much higher temperatures than at which they order). As you might have inferred from the above, I agree completely with the conclusions of the authors. I think this is an important discussion to have, and am pleased the authors have taken the time to write such a paper.

Authors: We thank the referee for his/her careful reading and valuable and positive comments on our manuscript.

2. Referee: that being said, it isn't clear to me that the "tone" of the paper is appropriate and/or the claims taken at face-value are necessarily correct. First, how is the key point of this paper different from a previous one of the authors? "Polymorphous band structure model of gapping in the antiferromagnetic and paramagnetic phases of the Mott insulators MnO, FeO, CoO, and NiO". Giancarlo Trimarchi, Zhi Wang, and Alex Zunger Phys. Rev. B 97, 035107 – Published 5 January 2018

Authors: We now list on pp. 5-6 previous DFT papers on this problem, including Trimarchi *et al*, and discuss the main differences:

*"There are certainly a number of papers over the years that have shown gapping in 3d oxides using appropriate DFT and taking into account the structural, electronic and magnetic degrees of freedom appearing in oxides (i.e allowing spin-polarization, lower energy phase searches for instance)²⁴⁻³². With the exception of Ref.²⁵, many of these studies focused on the low-temperature spin-ordered phase, even though the gapping usually appear in the paramagnetic (PM) spin-disordered phase. In Ref.²⁵, Trimarchi *et al* have proposed a strategy to model PM phases of simple binary oxides, identifying mechanisms to explain gapping in binaries such as MnO and NiO. The current paper focuses on ternary compounds such as ABO₃ that possess strongly entangled structural, electronic and magnetic degrees of freedom, yielding far more complex physical behaviour and large range of functionalities encompassing ferroelectricity, magnetism, thermoelectricity for instance. So far, DFT studies have not addressed the PM phases of 3d transition metal ABO₃ materials, nor elaborate the specific modalities of DFT*

required to produce gapping, nor have they systematically described the 'Periodic Table of gapping' by considering the whole range of trends for the chemical series ABO_3 with different A and B atoms. The continuing impression is that DFT itself is failing. It thus seems that the question of what is the minimal theory that describes the basic ground state phenomenology across the ABO_3 series — symmetry broken or symmetry conserving; statically correlated (mean-field treatment of electron-electron repulsion as in DFT) or dynamically correlated — is still unsettled."

3. Referee: i have two competing views of this paper:

Referee 1): this is an important paper that needs to be published in a journal such as Nat Comm,

Authors: This is our view. We will explain and reinforce this view in the revised text.

Referee: 2) this is an artificial problem in the sense that anyone who works on transition metal oxides, and knows what they are doing, already understands these issues. with regard to the latter, i think the authors need to do a much better job of citing the existing literature where researchers have pointed out similar conclusions as the authors here. I would like to emphasize that i think the present paper is still important, as nothing published was done as systematically nor was performed across the perovskite "period table."

Authors: We agree that it will be good to discuss this alternative view in the manuscript text, and use this opportunity to cite more previous papers on the subject and analyse the issues. We do this in two ways:

First, on pp. 5-6 we cite and discuss previously published papers by careful DFT practitioners on some isolated cases of 3d oxides (text given in response to comment 1):

"There are certainly a number of papers over the years that have shown gapping in 3d oxides using appropriate DFT and taking into account the structural, electronic and magnetic degrees of freedom appearing in oxides (i.e allowing spin-polarization, lower energy phase searches for instance)²⁴⁻³². With the exception of Ref.²⁵, many of these studies focused on the low-temperature spin-ordered AFM or FM phases rather than the all-important the paramagnetic spin-disordered phase. In Ref.²⁵, Trimarchi et al have proposed a strategy to model PM phases of simple binary oxides, identifying mechanisms explain gapping in binaries such as MnO and NiO. The current paper focuses on ternary compounds such as ABO_3 that possess strongly entangled structural, electronic and magnetic degrees of freedom, yielding far more complex physical behaviour and large range of functionalities encompassing ferroelectricity, magnetism, thermoelectricity for instance. So far, DFT studies have not addressed the PM phases of 3d transition metal ABO_3 materials, nor elaborate the specific modalities of DFT

required to produce gapping, nor have they systematically described the 'Periodic Table of gapping' by considering the whole range of trends for the chemical series ABO_3 with different A and B atoms."

Second, on p.5 we note that despite such discussions in the literature by rigorous DFT practitioners, the dominant tone in the oxide condensed matter theory literature is that of strongly correlated community, stating over and over that DFT cannot capture gapping in PM phase of perovskites, disqualifying such a method for many other types of studies on 3d oxides. We now add to the text on p.4 a number of quotes from the literature explaining that DFT fails in treating 3d oxides:

"Even though naïve DFT approximations correspond to extremely high-energy solutions, they were often used in the literature to suggest that DFT band theory fails to explain gapping of Mott insulators, the latter being argued to require instead an explicitly correlated approach¹⁷⁻²¹. In the specific case of rare-earth nickelates, it was stated that "Standard DFT and DFT+U methods fail to describe the phase diagram, with DFT predicting that all compounds remain metallic and undisproportionated.... These results establish that strong electronic correlations are crucial to structural phase stability and methods beyond DFT and DFT+U are required to properly describe them" in Ref.²², "spin polarized DFT shows metallic behaviour with neither magnetism nor bond disproportionation...: This qualitative structural errorsignals the importance of correlations" in Ref.²³ or "While density functional band theory (DFT) is the workhorse of materials science, it does not capture the physics of the Mott/charge-transfer insulator transition" in Ref.¹⁸. Moreover, gapping (and related structural distortions) in the paramagnetic phases of oxides is often claimed to be unreachable by DFT simulations: "However, these methods (i.e LDA,GGA,LDA+U) usually fail to describe the correct electronic and structural properties of electronically correlated paramagnetic materials" and "Therefore, LDA+U cannot describe the properties of $LaMnO_3$ at $T > T_N$ and, in particular, at room temperature, where $LaMnO_3$ is a correlated paramagnetic insulator with a robust JT distortion" in Ref.¹⁷ or "Although they cannot represent the paramagnetic insulating state, static mean field theories such as DFT, DFT+U, and hybrid functional approaches may capture some of the physics of the AFM insulating ground state" in Ref.²³."

Because of these criticisms we decide to study the reasons for the failure of naïve DFT. The present paper has two main points:

(1) we provide a concrete description of the specific computational modalities of DFT calculations that one needs to apply study gapping in ABO_3 perovskite oxides (these four "knobs" include octahedra rotations, orbital broken symmetries, SQS approach to PM phases, polymorphous representation with a unique local potential for each transition metal element);

(2) we show that all of the leading observed trends in spin ordered and spin disordered properties are explained by such correct DFT approach throughout the 3d series in ABO₃. These two innovations will make this paper highly cited for a long time.

4. Referee: The present paper focuses a lot on the "plus U" method in KS-DFT and how people state that since ks-dft fails one needs to use the plus U method to treat correlations. They correctly point out that it does not capture the physics of the Hubbard model. In my mind, the plus U method is best thought of as a poor-man self-interaction correction, and the authors even say as much.

Authors: Indeed, the plus U method is a (simplified) correction to the spurious self-interaction term rather than modelling Hubbard physics. We now emphasize it on pp.7-8:

"A number of calculations have used this 'DFT+U' approach, where DFT is amended by an on-site potential that removes part of the spurious self-interaction error and thereby creates a distinction between occupied and empty states producing at times gapped states^{25,27,28,30,32,39-42}. DFT+U successfully obtained gapping in simple binary oxides such as MnO, FeO, CoO or NiO²⁵, dioxides such as UO₂²⁶ and in the spin-ordered phases of the much more complex 3d transition metal ABO₃ compounds^{27-32,41,43-45}. These successes have, in part, propagated the view that it is the interelectronic repulsion akin to the Hubbard Hamiltonian that produces gapping in DFT+U. In fact, the role of the on-site potential U in DFT+U, where U is a one body on-site potential shifting the d orbital to deeper energies, is distinct from the Hubbard Hamiltonian, where it truly represents interelectronic repulsion. Furthermore, U in DFT is actually not required to produce gapping as illustrated by U-free calculations for spin-ordered state of several ABO₃ compounds^{28,46-53} as well as for other complex oxides such as VO₂²⁴ and La₂CuO₄⁵⁴ for instance. Further details on the DFT calculations and on the choice of U parameters are provided in Supplementary S4."

5. Referee: 3a. a central aspect of the authors argument is that since "DFT" and subsequently "plus U" is a single-determinate theory, it doesn't have correlation effects.

Authors: No. DFT has static correlation effects, but no dynamical effects. To be clear on this point and on a number of related referee questions (points 7,8,9,10,11 below) we now added to the introduction a general explanation of the terminology and viewpoints in the field, to the benefit of the broad readership of NCOMM. It is now stated for clarity on p. 6:

"Following the standard E. Wigner definition³³, correlations are considered to be all physical effects beyond mean field Hartree-Fock methods³⁴. An often voiced popular (but incomplete) analogous view applied to DFT is that correlation is everything that DFT does not

get right. According to this paradigm, the success of (a more general) DFT in describing the trends in the properties of ABO_3 d-electron perovskites, described in this paper, would suggest that whereas ABO_3 oxides may be (structurally and magnetically) complicated, they are not necessarily strongly correlated.

In fact, DFT is an exact formal theory for the ground-state properties for the exact exchange-correlation energy functional³⁵, so there is no reason in principle why the properties noted above couldn't be captured by the ultimate DFT. The paradigm that correlation is what DFT does not get right is, therefore, most likely, a diminishing domain. In the present paper we show that a single-determinant mean-field approach with static correlations such as density functional theory (DFT) successfully describes the basic properties across the 3d perovskite series of materials, including both spin-ordered LT and paramagnetic HT phases (Fig.2). As will be described below, achieving this requires (a) allowing sufficient structural flexibility (a polymorphous description) in the description of the various phases so that symmetry breaking reduced crystal symmetries that could lift degeneracies (octahedral rotations, Jahn-Teller and bond disproportionation effects) and electronic instabilities, could occur, should they lower the total energy, and (b) using exchange-correlation functional in the KS-DFT that distinguishes occupied from unoccupied states (such as DFT+U; Self interaction corrected functionals) and minimizes the delocalization error³⁶. We are not claiming that all current and future d or f-electron compounds can be fully explained by the properly executed DFT with currently known XC functionals, nor that properties other than the basic gapping, moments, and structural displacements can be always predicted. These are left for future research. Such failures, if and when found, would provide legitimate challenges for explicitly correlated methodologies to explain, rather than implying a universal role for symmetry-conserving dynamical correlations across the board. This is a significant result because it suggests a rather simple tool such as DFT (requiring a low computational effort with respect to heavier machineries treating dynamical correlations) offers a single platform to study reliably and with sufficient precision not only band gap formation, structure and magnetism in ABO_3 , but also – in the future – doping, defect physics and interface effects.”

6. Referee: BY this logic however, is it true that any model-Hamilton theory solved within mean field theory does not have correlations? I do understand that the proper definition of a "Correlated" system is one in which cant be described by a single determinate, however, i think the entire many-body community (for lack of a better way to describe them) would still refer to the system as correlated even if a mean field theory describes the physics. I think this is a language issue, not physics?

Authors: We prefer to refrain from commenting on the philosophy of Model Hamiltonian mean field approaches which is not the subject of our paper. In fact, we do not use the term “correlated materials” but just call ABO_3 “perovskite oxides with 3d atoms”. We let the reader

decide what level of theory is needed to address their basic properties. Our paper gives a clear answer to this question for the ABO_3 group.

7. Referee: 3b. I know the authors know this, but you consistently refer to DFT as a single determinate theory. Dont you mean Kohn-sham DFT?

Authors: Yes. We have now used the full title “Kohn-sham DFT” in the manuscript.

8. Referee: Also, please take a step back and reread your paper, there are several places where seem to suggest that DFT is not an exact theory of the many-body state. which brings me to a question or comment.

Authors: We agree. We are now making a distinction between formal Density Functional Theory with the perfect functional, and currently practiced DFT. This is now addressed by our added text given in our reply to comment 5 above. This text appearing in the Introduction should now give the proper, accurate tone to the whole paper.

9. Referee: I have to confess that i am not an expert on the theoretical structure of DFT itself, but my understanding is that even Kohn-Sham DFT is an exact many-body theory, we just don't know the Exchange Correlation functional.

Authors: Indeed. Please see our reply to comment 5.

10. Referee: So, then isnt it natural that a "better" XC-functional, contains a "better" description of correlations? So wrt the discussion of the meta-gga and band gaps, it isnt clear to me the meta-gga works "better" because of a "better" description of correlations. In fact, i think this is what the authors are saying indirectly?

Authors: We have now introduced a new title to our previous text, p 7-8, describing “*The DFT features that are needed beyond the N-DFT*”. This has two sub sections: (a) the DFT XC format used and (b) the polymorphous representation of structural degrees of freedom. Together, they tell the reader what should they do to practice a sufficiently general DFT format. This section is one of the most useful ones in this manuscript and is likely to become a highly cited reference for people wanting to do good quality DFT calculations on “correlated oxides”.

11. Referee: I have to say, it seems this paper was written to "preach" to one community, the community who loudly mistakenly claim "DFT" does not describe correlations (this is what I meant by "tone" in my first paragraph). In doing so, those who actually know better can misread some of the statements in the paper. (I again want to state, I know the authors are in the camp who "know better", but I think they would benefit if they "take a step back" and think about who will read this paper.)

Authors: Point taken. We are now even more careful than in the original manuscript in not giving the impression that all DFT practitioners were naïve... We have modified our paper accordingly and clearly state that properly executed DFT reasonably give all ground state properties, including trends with respect to the *d* filling.

We now list on p. 5 and p. 7 previous DFT papers on this problem and discuss the main differences reply to comment 2:

"There are certainly a number of papers over the years that have shown gapping in 3d oxides using appropriate DFT and taking into account the structural, electronic and magnetic degrees of freedom appearing in oxides (i.e allowing spin-polarization, lower energy phase searches for instance)²⁴⁻³². With the exception of Ref.²⁵, many of these studies focused on the low-temperature spin-ordered phase, even though the gapping usually appear in the paramagnetic (PM) spin-disordered phase. In Ref.²⁵, Trimarchi et al have proposed a strategy to model PM phases of simple binary oxides, identifying mechanisms to explain gapping in binaries such as MnO and NiO. The current paper focuses on ternary compounds such as ABO₃ that possess strongly entangled structural, electronic and magnetic degrees of freedom, yielding far more complex physical behaviour and large range of functionalities encompassing ferroelectricity, magnetism, thermoelectricity for instance. So far, DFT studies have not addressed the PM phases of 3d transition metal ABO₃ materials, nor elaborate the specific modalities of DFT required to produce gapping, nor have they systematically described the 'Periodic Table of gapping' by considering the whole range of trends for the chemical series ABO₃ with different A and B atoms.

*"A number of calculations have used this 'DFT+U' approach, where DFT is amended by an on-site potential that removes part of the spurious self-interaction error and thereby creates a distinction between occupied and empty states producing at times gapped states^{25,27,28,30,32,39-42}. DFT+U successfully obtained gapping in simple binary oxides such as MnO, FeO, CoO or NiO²⁵, dioxides such as UO₂²⁶ and in the spin-ordered phases of the much more complex 3d transition metal ABO₃ compounds^{27-32,41,43-45}. These successes have, in part, propagated the view that it is the interelectronic repulsion akin to the Hubbard Hamiltonian that produces gapping in DFT+U. In fact, the role of the on-site potential *U* in DFT+U, where *U* is a one body on-site potential shifting the *d* orbital to deeper energies, is distinct from the Hubbard Hamiltonian, where it truly represents interelectronic repulsion. Furthermore, *U* in DFT is actually not required to produce gapping as illustrated by *U*-free calculations for spin-ordered*

state of several ABO_3 compounds^{28,46–53} as well as for other complex oxides such as VO_2 ²⁴ and La_2CuO_4 ⁵⁴ for instance.”

Author remarks on Reviewer #2 :

Referee: The authors systematically studied the magnetic, electronic, and structural properties of a set of prototypical ABO_3 perovskites, and found that within DFT augmented by the empirical U to correct the intrinsic self-interaction error all the properties can be predicted to be consistent with experimental results, including the paramagnetic phases under the SQS model, via symmetry-breaking energy-lowering procedures. In particular, all insulators were predicted to have gaps. To eliminate the complication caused by U, they applied the SCAN density functional to those insulating phases, and all predictions for insulators remained valid. The authors therefore concluded that whereas ABO_3 oxides may be complicated, they are not necessarily strongly correlated. As these compounds are typically regarded as strongly correlated materials, where strong inter-electron interaction is thought to be critical and computationally efficient conventional density functionals not applicable, this study will have a significant impact on studies of traditionally defined strongly correlated materials. The results the authors presented are solid and the analysis of gapping mechanism should be interesting to a wide audience. I therefore recommend the manuscript for publication in Nat. Comm.

Authors: We thank the referee for his very positive comments on our study.

Referee: The following are several suggestions:

1.Referee 1) The authors should discuss how well the SQS model models the PM phases since this is one of the critical points in this manuscript.

Authors: We thank the referee for raising that point. We have already included in the originally submitted manuscript the convergence tests with respect to SQS size in the supplementary material. We found that a 160 atoms unit cell was providing satisfying convergence on the energy, magnetic moments and band gaps of two perovskites, namely $YTiO_3$ (d^1) and $YNiO_3$ (d^7). We now point more directly in the main text on p.8 to the supplementary explaining briefly what is discussed there.

“We use the Special Quasi Random (SQS)⁵⁵ construct that selects supercells of given crystalline space group symmetry such that the occupation of sites by spins follows a random pair and multi body correlation functions (appropriate to the HT limit) with a total moment of zero. The method is described in details in Refs.^{25,55}. We have tested the convergence of results with

respect to energy, magnetic moment and band gaps on two “limit” compounds (namely YTiO_3 , i.e. d^1 , and YNiO_3 , i.e. d^7). We have found that a 160 atoms unit cell is sufficiently large to produce converged results (see Supplementary S5 for details on SQS and the generated supercells). This polymorphous representation²⁵ provides an opportunity to break spatial symmetry, should the total energy be lowered in doing so.”

2. Referee: 2) In the occupation number fluctuations, the method to lift the degeneracy by nudging the occupation, the authors should discuss about the metastable states encountered in DFT+U, as discussed in B. Meredig, et al, PRB, 82, 195128 (2010). This problem makes DFT+U less ideal for this study, which however has been alleviated by the application of SCAN on insulators.

Authors: This is now addressed on p. 9.

“The problem of finding the site occupations that lead to minimum energy is, in general, a non-trivial optimization problem²⁶, especially when done for the artificial case of (i) high symmetry unit cell and (ii) rigid lattice. However, here we (i) use a supercell that already has low symmetry (3d atoms have their own local environment) and (ii) allow the lattice to respond to site fluctuations in occupation numbers. Additionally, as we will show below, most of the structural distortions (O_6 rotations and anti-polar motions of ions) appearing in the ABO_3 perovskites mix the “cubic” orbitals removing orbital degeneracies present in the initial cubic phase. Thus, the existence of meta-stable phases in the cubic cell is not “so crucial” for our study and these orbital broken symmetries are there to probe the mechanism yielding the band gap (Jahn-Teller motion or disproportionation for instance). We have tested few initial guesses and only the most stable phase reached after the self-consistency is kept. Finally, for disproportionating materials (e.g. YNiO_3 or CaFeO_3), different types of initial “nudging” were performed such as (1,0) e_g partners occupancies on all B sites or (1,1) and (0,0) e_g partners occupancies between neighbouring B sites.”

3. Referee: 3) For this work, when citing SCAN, it is appropriate to also cite J. Furness, et al, Communication Physics, 1, 11 (2018), where SCAN is demonstrated to open the gap for the pristine La_2CuO_4 .

Authors: We apologize for missing to cite this key reference. It is now added to the manuscript.

4. Referee: 4) At page 6, use (1), (2), and (3) instead of (a), (b), and (c) for the symmetry breaking channels.

Authors: Done.

Reviewers' comments:

Reviewer #1 (Remarks to the Author):

I have read the responses of the authors and reread the manuscript. I am satisfied and convinced by the authors responses. The manuscript should be published in Natt Comm.

Reviewer #4 (Remarks to the Author):

The authors presented a systematic study on magnetic, electronic and structural properties in low-temperature (spin-ordered) and high-temperature (paramagnetic) phases of perovskite oxides. A polymorphous DFT method with a flexibility to produce symmetry-breaking patterns is employed. The experimental metal-nonmetal behavior and magnetism is well reproduced by the single-determinant DFT method, leading the authors to conclude that ABO₃ perovskites are not necessary strongly correlated, in contrast to a common view that these materials are strongly correlated with the Mott-Hubbard physics. As stated in well-motivated introduction, the authors tackled an important open problem in condensed-matter physics and the results will encourage future DFT studies for functional perovskites. Computational results are reliable within the present approximation.

I have following criticisms and suggestions for a better visualization.

1) The authors referred to DFT+DMFT studies indicating the importance of the dynamical electronic correlation for low-energy physics including gap opening. Opposed to these studies, the authors show the gap is obtained by the single-determinant DFT method by allowing symmetry-breaking patterns, that is the base of their conclusion that "the perovskites are not necessary strongly correlated" (in Page.6) and "they are not obviously strongly dynamically correlated materials" (in Page.22). However, can one derive the strong conclusion only from the gapped-non-gapped behavior? As explained in the author's introduction, the correlations are usually defined as all physical effects beyond static mean-field methods. Indeed, not only the gap but also e.g. low-energy (mass) renormalization is a widely-accepted "basic" property of electronic correlation. As an example, SrVO₃, a studied compound, shows a correlated behavior beyond HF descriptions, i.e. a formation of Hubbard bands and a coherent peak at EF due to the electronic correlation in the ground state, revealed by photoemission experiments, which classifies SrVO₃ into strongly-correlated-metal category. I think the present single-determinant theory cannot describe it and dynamical correction in the ground-state description is surely necessary to describe the basic property of the correlated perovskites. So what the authors can derive from the present resents is not such a general statement about the electronic correlation but just fact that the paramagnetic gap can be simulated within the single-determinant theory. So the authors should reconsider these conclusions exaggerated beyond the present results.

2) The authors referred to previous studies (P.5) reporting the failure of the (naive) DFT, e.g. "While DFT is the workhorse of materials science, it does not capture the physics of the Mott/charge-transfer insulator transition". I admit the paramagnetic gap in the present study and in Trimarchi et al is interesting. However, the present study cannot rule out the conventional Mott-Hubbard (MH) mechanism as the leading mechanism of the insulating gap in the studied transition-metal oxides. In previous (P.5) and other studies, the MH mechanism is shown to open the gap. Though various symmetry-breaking are studied here, the comparison taking the Mott-Hubbard mechanism into account is not performed. So the present study just proposes another possible explanation (gap without the Mott mechanism) but I believe the triggering mechanism of the gap remains as an open problem. The authors should comment on this point. Currently I am taking sides with the Mott-Hubbard scenario in previous studies, most of them with the DFT+DMFT method, since they have shown satisfactory agreement with experiments, e.g. photoemission, a

direct probe of the insulating-gap “character” (e.g. Mott or charge-transfer type) in a wide of transition-metal oxides including perovskites, that is not given in the authors’ scenario and in the present manuscript.

3) The authors discussed the insulating behavior of YNiO₃. By referring to DFT+DMFT study by Park et al, the authors criticized that the charge disproportionation (CD) is entirely predictable by the static mean-field theory. However does the present method describe the correct physics? The DFT+DMFT study pointed out that the CT between Ni_L and Ni_S sites does not accompany the insulating gap (similar conclusion in Ref. 66), while the authors found a clear CD between the Ni sites. My doubt is that the insulating gap in the present result is just an artifact due to an overestimation of CD, expected in a static mean-field treatment. There is no reason to believe the present DFT+U result over the DFT+DMFT one. So I ask the authors to give more quantitative discussion about CD with a comparison to the DFT+DMFT study. Readers including me will not be satisfied by the rough statement in page.20, “physical charge density around Ni_S and Ni_L are nearly indistinguishable” since the charge density at Ni_S and Ni_L sites looks different substantially.

4) Can authors give the definition of the averaged PDOS in the paramagnetic phase? To be more specific, please explain (in caption) what the upper and lower panels in Fig.3b represent. Then how does the orbital- (and element-) resolved total density of states over the supercell look like?

Reviewer #1 (Remarks to the Author):

Referee: I have read the responses of the authors and reread the manuscript. I am satisfied and convinced by the authors responses. The manuscript should be published in Nat. Comm.

Authors: We thank the referee for his/her positive comments.

Reviewer #4 (Remarks to the Author):

(1) Referee: The authors presented a systematic study on magnetic, electronic and structural properties in low-temperature (spin-ordered) and high-temperature (paramagnetic) phases of perovskite oxides. A polymorphous DFT method with a flexibility to produce symmetry-breaking patterns is employed. The experimental metal-nonmetal behavior and magnetism is well reproduced by the single-determinant DFT method, leading the authors to conclude that ABO₃ perovskites are not necessarily strongly correlated, in contrast to a common view that these materials are strongly correlated with the Mott-Hubbard physics. As stated in well-motivated introduction, the authors tackled an important open problem in condensed-matter physics and the results will encourage future DFT studies for functional perovskites. Computational results are reliable within the present approximation.

Authors: We thank the referee for his/her positive remarks on our work, especially for stating that “Computational results are reliable within the present approximation”

(2) Referee: I have following criticisms and suggestions for a better visualization.

The authors referred to DFT+DMFT studies indicating the importance of the dynamical electronic correlation for low-energy physics including gap opening. Opposed to these studies, the authors show the gap is obtained by the single-determinant DFT method by allowing symmetry-breaking patterns, that is the base of their conclusion that “the perovskites are not necessarily strongly correlated” (in Page.6) and “they are not obviously strongly dynamically correlated materials” (in Page.22). However, can one derive the strong conclusion only from the gapped-non-gapped behavior?

Authors: Our conclusion that “*perovskites are not necessarily strongly dynamically correlated*” is not based only on the gapped-non-gapped behavior, but on the reproduction of a list of other ground state properties. To make sure that the reader does not miss this essential point we now re-iterate the central result in the Abstract, stating:

“Here, we show that contrary to such previous experience that was based on simplified and restricted versions of DFT, if one allows symmetry-breaking energy-lowering crystal symmetry reductions and electronic instabilities within DFT, one successfully and systematically recovers the observed leading metal-non-metal features, magnetic moments, type of magnetic and crystallographic ground state, trends in bond disproportionation and ligand hole effects, Mott versus charge transfer insulator behaviors, and the amplitude of structural deformation modes including Jahn-Teller. We show that this is the case for the whole ABO_3 perovskite oxide series, within their low temperature spin-ordered and high temperature disordered paramagnetic phases and provide a classification of the four-triggering mechanism of the gap formation.”

What makes this result noteworthy is that the literature includes a large number of statements that DFT fails, many such statements are cited verbatim on p. 5. We feel it is time to set the record straight by performing careful and systematic calculations. There can be no doubt that our results are not focused only on the gapped-non-gapped behavior, as the referee thought.

(3) Referee: As explained in the author’s introduction, the correlations are usually defined as all physical effects beyond static mean-field methods. Indeed, not only the gap but also e.g. low-energy (mass) renormalization is a widely-accepted “basic” property of electronic correlation. As an example, $SrVO_3$, a studied compound, shows a correlated behavior beyond HF descriptions, i.e. a formation of Hubbard bands and a coherent peak at E_F due to the electronic correlation in the ground state, revealed by photoemission experiments, which classifies $SrVO_3$ into strongly-correlated-metal category. I think the present single-determinant theory cannot describe it and dynamical correction in the ground-state description is surely necessary to describe the basic property of the correlated perovskites. So what the authors can derive from the present results is not such a general statement about the electronic correlation but just fact that the paramagnetic gap can be simulated within the single-determinant theory. So the authors should reconsider these conclusions exaggerated beyond the present results.

Authors: We are happy that the referee brought the example of $SrVO_3$ saying that “the present single-determinant theory (i.e. DFT) cannot describe it (i.e. $SrVO_3$) and dynamical correction in the ground-state description is surely necessary to describe the basic property of the correlated perovskites”. The referee indeed believes this because he/she has not seen reliable DFT calculations. Thus, we now add to the paper in the section describing $SrVO_3$ (p. 16 the following text:

“It is often thought that SrVO₃ is a good example of failure of mean-field like theories, and that SrVO₃ shows a correlated behavior beyond HF descriptions, i.e. formation of Hubbard bands and a coherent peak at E_F due to the electronic correlation in the ground state, revealed by photoemission experiments. This viewpoint classifies SrVO₃ into strongly-correlated-metal category and dynamical correction in the ground-state description is surely necessary to describe the basic property of the correlated perovskites. We emphasize that the metallic nature of SrVO₃ is a simple consequence of the cubic structure being the lowest total energy phase. We note that the SQS-PM approach that is deliberately restricted to single determinant mean field gives a clear d band (called in Hubbard models as upper Hubbard band) and this band width W associated with t_{2g} is 1.6 eV (SQS-PM DFT), matching well the Angle Resolved Photo Emission Spectroscopy⁶³ (ARPES) that gives W_{t_{2g}}= 1.3 eV and DMFT⁶⁴⁻⁶⁶ in which W_{t_{2g}} is estimated to 0.9, 1.2 or 1.35 eV depending on the DMFT parameters. In contrast, N-DFT using a single monomorphous unit cell gives a band width W_{t_{2g}} of 2.6 eV as cited by Ref⁶⁶.”

We remind the referee and readers that DFT does not provide quantitatively accurate gaps, so we state on p. 10 line 5:

“We note that just as is the case in the highly uncorrelated compounds Si and GaAs, DFT often does not give accurate band gaps (and hence, effective masses), and a GW correction is needed⁵⁶. The 3d oxides ABO₃ are no exception; we expect that more quantitative gap values and (renormalized) masses will be available as GW is applied to our SQS-DFT.”

(4) Referee: The authors referred to previous studies (P.5) reporting the failure of the (naive) DFT, e.g. “While DFT is the workhorse of materials science, it does not capture the physics of the Mott/charge-transfer insulator transition”. I admit the paramagnetic gap in the present study and in Trimarchi et al is interesting. However, the present study cannot rule out the conventional Mott-Hubbard (MH) mechanism as the leading mechanism of the insulating gap in the studied transition-metal oxides. In previous (P.5) and other studies, the MH mechanism is shown to open the gap. Though various symmetry-breaking are studied here, the comparison taking the Mott-Hubbard mechanism into account is not performed.

Authors: The correct historical timeline has been as follows:

First, DMFT folks have shown, using mostly N-DFT calculations, that DFT fails. To make sure the reader remembers this we have collected on p.5 explicit citations from DMFT papers stating just this.

Second, DMFT folks have advanced a complex viewpoint of developing DMFT codes and showing in just a few cases of ABO₃ compounds that DMFT works. But note that in most of these calculations they did not have a total energy capability so they did not predict crystal structures but rather copied them from experiment. Thus, these simulations are absolutely not a proof of the dominant role of dynamical electron correlations over structural symmetry

breaking modes for the gap opening. All DFMT studies on ABO_3 materials suggesting the importance of the Mott-Hubbard mechanism used the experimentally observed structure in the insulating phase. So, symmetry breaking modes are already there. What they do is just modelling electron interactions using the extreme picture of the Mott-Hubbard physics, but they do not disentangle the role of the lattice over dynamical correlations.

Third, we suggested in the current paper that we wish to find the minimal physical picture that reproduces the trends in the ground state properties across the full ABO_3 series. We find that the single determinant mean field does this. We show in our manuscript that a simpler level of description of electron interactions with no dynamical correlations is largely sufficient to describe perovskites properties (gap, magnetic moments, structural symmetry breaking modes...) in very good agreement with experiments. We are not ruling out any non-minimalistic high order effects.

To make sure that the reader gets this we now add to p.3, bottom, the text:

“Note that in most dynamically correlated calculations the crystal structure and its subtle distortions were not predicted from the underlying electronic structure (with the exception of the works of Marianetti et al¹⁷ and Leonov et al¹⁸ for instance) but rather copied from experimentally observed structures. Thus, these simulations are not a proof of the dominant role of dynamical electron correlations over structural symmetry breaking modes for the gap opening. This question is open.”

We also add to the end of the section (p.7) on the alleged failures of DFT:

“We find indeed that the ground state properties of 3d ABO_3 oxides are not good examples of the failure of DFT or the need for special effects outside DFT.”

(5) Referee: So the present study just proposes another possible explanation (gap without the Mott mechanism) but I believe the triggering mechanism of the gap remains as an open problem. The authors should comment on this point.

Authors: Section IV. Called **“Gapping Mechanisms”** on p.12-23 (note how explicit it is!) describes in great detail the triggering mechanism of the gap. Please read this again and you will see that without a doubt the *triggering mechanism of the gap formation* (we now borrowed this term and use it on p. 12, thank you) is fully described. We now add to the **abstract**, to make sure no one misses this: *“We provide a classification of the four triggering mechanisms of the gap formation”*.

(6) Referee: Currently I am taking sides with the Mott-Hubbard scenario in previous studies, most of them with the DFT+DMFT method, since they have shown satisfactory agreement with experiments, e.g. photoemission, a direct probe of the insulating-gap “character” (e.g. Mott or charge-transfer type) in a wide of transition-metal oxides including perovskites, that is not given in the authors’ scenario and in the present manuscript.

Authors: Actually, our DFT+U correctly predict band positions: d^1 (RTiO₃), d^2 (RVO₃) and d^4 (LaMnO₃) compounds have d like band edges (usually classified as Mott insulators) while other materials (CaMnO₃, LaFeO₃, CaFeO₃, YNiO₃...) have p like conduction band edge (usually classified as charge transfer insulators). We invite the referee to look at the projected density of states reported in Figures 3, 4, 5 and 6 and in the Supplementary Material showing the dominant states at the top (bottom) of the valence (conduction) band. Band positions just depend on the d filling and on the formal oxidation states of the compounds, this is not a dynamical correlation effect. Thus, compounds with d like band edges are not necessarily Mott Insulators (gapped because of strong correlations) but just have certain orbital character that happens to be d or p .

(7) Referee: The authors discussed the insulating behavior of YNiO₃. By referring to DFT+DMFT study by Park et al, the authors criticized that the charge disproportionation (CD) is entirely predictable by the static mean-field theory. However does the present method describe the correct physics? The DFT+DMFT study pointed out that the CT between Ni(L) and Ni(S) sites does not accompany the insulating gap (similar conclusion in Ref. 66), while the authors found a clear CD between the Ni sites. Readers including me will not be satisfied by the rough statement in page.20, “physical charge density around Ni(S) and Ni(L) are nearly indistinguishable” since the charge density at Ni(S) and Ni(L) sites looks different substantially.

Authors: Charge disproportionation was discussed in greater detail in our previous papers and in the present paper in the section on YNiO₃ we do not dwell on this. The papers are: “Complete phase diagrams of rare-earth nickelates from first-principles”, npj Quant. Mater. **2**, 21 (2017) and “Bond disproportionation, charge self-regulation, and ligand holes in s - p and in d -electron ABX_3 perovskites by density functional theory” Phys. Rev. B **98**, 075135 (2018). We clearly show in these papers, which are not the subject of the current review, that the integral of the electronic density over a sphere centered around Ni cations is virtually the same for both Ni sites and corresponds to Ni charges that hardly change at the metal-insulator transition. Please see Fig 3 in Phys. Rev. B **98**, 075135 (2018). Thus, our DFT simulations give conclusions similar to those drawn on the basis of DMFT: there is an absence of “physical charge ordering”. Proper references to previous work was explicitly written in the manuscript “although charge disproportionation occurs in the “ $3d + ligand$ ” unit cell, the physical charge density $\rho(\vec{r})$ around Ni_s and Ni_l cations are nearly indistinguishable, owing to a charge self-

regulation mechanism⁶³⁻⁶⁵, whereby the ligands resupply the 3d site with charge lost. This result has sometimes been interpreted in terms of a 'ligand-hole' language^{23,28,66} (p21). We now added for clarity: "There is no physical charge ordering" to the end of this paragraph and we have rewritten the text on p. 20-24 to remove any ambiguity.

(8) Referee: My doubt is that the insulating gap in the present result is just an artifact due to an overestimation of CD, expected in a static mean-field treatment. There is no reason to believe the present DFT+U result over the DFT+DMFT one. So I ask the authors to give more quantitative discussion about CD with a comparison to the DFT+DMFT study.

Authors: To be clear on this we now add to the manuscript the text on p. 21 bottom and p. 22 top:

"That the extent of charge transfer, the lack of physical charge ordering, and the magnitude of disproportionation are reasonably depicted by DFT can be judged by examining the ensuing calculated bond lengths vs experiment: Table SI7 presented in the Supplementary S6 shows that the predicted amplitude associated with the bond disproportionation distortion is in excellent agreement with experimental quantities in CaFeO₃ and YNiO₃. Similarly, on different cases of disproportionation, Fig.4 of Ref⁶⁹ shows a similarly excellent agreement with the data."

We further note on P. 23 top:

"Finally, the disproportionation effects was described in detail by Park, Marianetti and Millis¹⁷ in terms of a "novel correlation effect", but as seen here, and in our Refs^{28,69}, it is entirely predictable by static mean field DFT. The very recent DMFT simulations of RNiO₃ compounds performed by Hampel et al⁷⁵ very closely reproduce our DFT+U results presented in Fig.6.f and 6g: (i) presence of an electronic instability toward a disproportionation of Ni³⁺ to Ni²⁺ and Ni⁴⁺ (ii) that is strongly coupled to the amplitude of rotations. As these DMFT results suggest nothing new relative to DFT; one would conclude that no special effects above DFT are needed to explain these facts."

(9) Referee: Can authors give the definition of the averaged PDOS in the paramagnetic phase? To be more specific, please explain (in caption) what the upper and lower panels in Fig.3b represent. Then how does the orbital- (and element-) resolved total density of states over the supercell look like?

Authors: We apologize for a mistake in the caption of figure 3 and we thank the referee for raising it. We now state in caption of figure 3: "Averaged projected density of states on B d

levels (grey) and O p levels (red) in CaMnO_3 (a) and LaFeO_3 (b) in a hypothetical cubic phase within the PM order. The averaged density of states is extracted by summing all contributions coming from all atoms in the PM cells, in each spin channels, for p (O, in red) and d (B, in grey) orbitals. Positive (negative) values stand for spin up (down). Orbital resolved DOS on t_{2g} (blue) and e_g (green) level is also provided, but for a given B site cation in the supercell.”